# MANY-OBJECTIVE MULTI-SOLUTION TRANSPORT

**Ziyue Li**[1]    **Tian Li**[2]    **Virginia Smith**[3]    **Jeff Bilmes**[4]    **Tianyi Zhou**[1*]
[1]University of Maryland    [2]University of Chicago    [3]Carnegie Mellon University
[4]University of Washington, Seattle
`{ziyueli, tianyi}@umd.edu, litian@uchicago.edu,`
`smithv@cmu.edu, bilmes@uw.edu`
Project: `https://github.com/tianyi-lab/MosT`

## ABSTRACT

Optimizing the performance of many objectives (instantiated by tasks or clients) jointly with a few Pareto stationary solutions (models) is critical in machine learning. However, previous multi-objective optimization methods often focus on a few objectives and cannot scale to many objectives that outnumber the solutions, leading to either subpar performance or ignored objectives. We introduce "**M**any-**o**bjective multi-**s**olution **T**ransport (MosT)", a framework that finds multiple diverse solutions in the Pareto front of many objectives. Our insight is to seek multiple solutions, each performing as a domain expert and focusing on a specific subset of objectives while collectively covering all of them. MosT formulates the problem as a bi-level optimization of weighted objectives for each solution, where the weights are defined by an optimal transport between objectives and solutions. Our algorithm ensures convergence to Pareto stationary solutions for complementary subsets of objectives. On a range of applications in federated learning, multi-task learning, and mixture-of-prompt learning for LLMs, MosT distinctly outperforms strong baselines, delivering high-quality, diverse solutions that profile the entire Pareto frontier, thus ensuring balanced trade-offs across many objectives.

## 1 INTRODUCTION

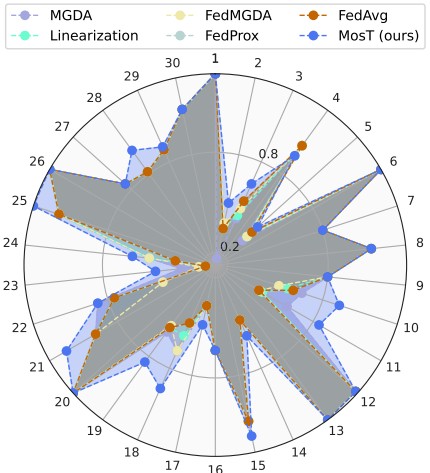

Figure 1: Accuracies of different methods outputting 5 solutions serving 30 objectives (clients) in federated learning. MosT results in a better coverage of all the objectives than the other baselines.

The underlying goal of many machine learning problems is to simultaneously optimize multiple objectives. Usually, no single solution (or model) is optimal for all objectives at once. Multi-objective optimization (MOO) aims to find a solution on the Pareto frontier where no objective can be improved without degrading others. One approach is to optimize a linear combination of all objectives, which may collapse to solutions for a small subset of objectives. Another method, multi-gradient descent algorithm (MGDA) (Désidéri, 2012) finds a common descent direction at each iteration to update the model so that no objective degrades. However, the trade-offs provided by MGDA solutions are not fully controllable even with recent advances like reference vectors to guide the search space among all the objectives (Mahapatra & Rajan, 2020), especially when the Pareto frontier is unknown and complicated (e.g., non-smooth or discontinuous).

Moreover, MOO approaches typically focus on two or three objectives and hardly scale to many objectives, as shown in Figure 1. As objectives increase, it is less plausible that they will reach an agreement on a single solution. Instead of balancing all of them, it is more appealing to find multiple diverse yet complementary solutions on the Pareto frontier each focusing

---

*Corresponding author.

on a local domain of objectives. This problem of finding $m$ Pareto solutions (or training $m$ models) for $n$ objectives can be understood as a multi-solution extension of MOO or a mixture of experts (MoE) (Jacobs et al., 1991) for multiple objectives. However, as $n$ increases, existing methods (e.g., those based on reference vectors or a uniform exploration of the Pareto frontier) can be computationally prohibitive and thus practically infeasible. It is also challenging to pre-determine the search regions of a few representative solutions profiling the entire Pareto frontier. The setting of $n \gg m$ is emerging in a variety of machine learning problems that involve many (i.e., big-$n$) users, domains, or evaluation criteria, each associated with a different training objective, but the total available data or computation can only support the training of $m \ll n$ models.

In this paper, we ask: *can we develop a solution-objective matching mechanism to guide the exploration of $m$ solutions on the high-dimensional ($n \gg m$) Pareto frontier?* For example, some objectives share similar structures; so optimizing them with the same model can bring common improvement, while optimizing separate models for objectives with mutual conflicts can effectively avoid poor performance and the tug-of-war among them. Hence, a matching between models and objectives after every model update step is able to explore the correlation among $n$ objectives along the optimization trajectory. Specifically, we capture the matching relations with a weight matrix $\Gamma \in \mathbb{R}_+^{n \times m}$ where $\Gamma_{.,j}$ reweighs the $n$ objectives that model $j \in [m]$ aims to optimize. For model $j$, such a *reweighted* single-model multi-objective optimization problem steers towards a domain expert focusing on a locally consistent subset of objectives. With complementary (i.e., every objective being equally covered) and balanced (i.e., no model dominating on most objectives) $\Gamma$'s to adjust the descent directions per iteration, we aim to find $m$ diverse yet complementary solutions that lie in the Pareto front.

More concretely, we model the optimization of $\Gamma$ as an *optimal transport* (OT) between the $m$ models and $n$ objectives, with two marginal constraints $\Gamma \mathbf{1}_m = \alpha$[1] and $\Gamma^\top \mathbf{1}_n = \beta$ in OT, which allow us to control the ratio of $n$ objectives assigned to each model and the ratio of $m$ solutions optimized for each objective. For example, with a uniform $\alpha = (1/n)\mathbf{1}_n$, the $m \ll n$ models are enforced by OT weights $\Gamma$ to focus on different subsets of objectives—otherwise, violate the $\Gamma \mathbf{1}_m = (1/n)\mathbf{1}_n$ constraint could leave some objectives uncovered. Similarly, with a uniform $\beta = (1/m)\mathbf{1}_m$, the training loads for the $m$ models tend to be balanced so no model would dominate the others on a majority of the objectives (Figure 1).

We propose an efficient algorithm for the above "*Many-objective multi-solution Transport* (**MosT**)" problem, which is a bi-level optimization between model parameters and objective-solution matching parameterized by $\Gamma$. Specifically, the upper-level is a $\Gamma$-reweighted MGDA problem for the $m$ models. The lower-level is a classic OT problem optimizing $\Gamma$ with marginal constraints. We further introduce a regularization term on top of $\Gamma$ to promote diversity of the optimal transport. Our algorithm converges to stationary points in the non-convex case, and converges to Pareto stationary solutions in the strongly-convex case under additional stability assumptions. We additionally extend MosT to handle the $n \ll m$ case by augmenting the $n$ objectives with random linear combinations of them ("**MosT-E**"). In practice, we introduce a curriculum for better model specialization by gradually varying the marginals $\alpha$ and $\beta$ in MosT—focusing more on 'models selecting objectives' at the earlier stage and later on transits to 'objectives selecting the best models'.

We apply MosT to various $n \gg m$ machine learning applications, spanning federated learning (McMahan et al., 2017), multi-task learning (Lin et al., 2019), and mixture-of-prompt learning (Qin & Eisner, 2021). Though the focus of this work is to address the challenges associated with scaling MOO to a large number of objectives, we also apply MosT to the $n \ll m$ scenarios, including fairness-accuracy trade-offs and other classic MOO problems (Appendix A.3). In all applications (Section 6), MosT finds diverse high-quality solutions on the Pareto front, consistently outperforming various strong baselines in terms of average accuracy and other popular metrics on the quality of multiple solutions, without extra computation cost.

## 2    RELATED WORK

**Single-Solution MOO.** The classic goal of MOO is to find some solution lying on the Pareto front of multiple objectives (Désidéri, 2012; Roy et al., 2023; Halffmann et al., 2022; Miettinen, 1999). One approach is to solve a linearized aggregation (i.e., weighted average) of all objectives. However,

---

[1] $\mathbf{1}_m$ denotes the $m$-dimensional all-ones vector and $\Gamma \mathbf{1}_m$ computes the row-wise sum of entries in $\Gamma$.

linearization, despite covering broad objective weights, may produce solutions in a small Pareto front area (Boyd & Vandenberghe, 2004). The multiple-gradient descent algorithm (MGDA) (Désidéri, 2012) is widely used due to its capability to handle complicated Pareto fronts and compatibility with gradient-based optimization. In this work, we use MGDA-style optimization methods in our algorithm, and compare with linearization-based objectives (with different weights) empirically (Section 6), showing that our approach can find more diverse Pareto stationary solutions.

**Multi-Solution MOO.** One line of work that aims to discover diverse solutions across the entire Pareto front builds upon MGDA and guides the search process via constraints of preference vectors (Lin et al., 2019; Mahapatra & Rajan, 2020) or constraints of other objectives (Zafar et al., 2017). These methods do not generalize well to the setting where there are many objectives (constraints) or the model dimension is large, since the number of preference vectors to explore the whole Pareto front may depend exponentially on these factors in the worst scenario (Emmerich & Deutz, 2018). Even in the setting where there are only a few (e.g., two or three) objectives, diversity of the preference vectors in the action space (during exploration) may not translate to diversity in the solution space. We showcase the superiority of MosT relative to these works in Section 6. Some works that balance Pareto optimality and solution diversity cannot guarantee the final solutions are on the Pareto front (Liu et al., 2021). For gradient-free methods, evolution strategies or Bayesian optimization (Coello, 2006; Sindhya et al., 2012) has been explored to find multiple (as opposed to one) Pareto stationary solutions. However, they are usually not efficient when solving practical MOO problems in machine learning due to the lack of gradient information (Liu et al., 2021; Momma et al., 2022); hence, we do not compare with those methods.

**Applications in Machine Learning.** We demonstrate MosT's effectiveness on applications including cross-device federated learning (McMahan et al., 2017). There is extensive prior research on personalized federated learning (e.g., Smith et al., 2017; Ghosh et al., 2020; Wu et al., 2022), i.e., outputting multiple related models instead of one, to serve all clients. Our approach can be viewed as a personalization objective here. Note that our goal is not to achieve the highest average accuracy for federated learning, but rather, *explicitly* balance multiple objectives and guarantee that all output solutions are Pareto stationary. We also explore multi-task learning (Lin et al., 2019) and mixture-of-prompt learning (Qin & Eisner, 2021) on standard benchmarks where each objective is a task or a training instance. For the $n \ll m$ case, following the setup in prior MOO works (e.g., Zitzler et al., 2000; Mahapatra & Rajan, 2020), we apply MosT to a toy problem and address (algorithmic) fairness/utility trade-offs (two objectives).

## 3 MosT: MANY-OBJECTIVE MULTI-SOLUTION TRANSPORT

In this section, we introduce MosT objective and the alternating optimization algorithm to solve it.

Let $L_i(\cdot)$ ($i \in [n]$) denote the empirical loss function of the $i$-th objective. When the number of objectives $n$ is much larger than the number of solutions $m$, it is possible that the learnt solutions (for example, simply by running MGDA for $m$ times with different randomness) cannot cover representative regions on the Pareto front. To address this, we use an assignment matrix $\Gamma \in \mathbb{R}_+^{n \times m}$ with constraints $\Gamma \mathbf{1}_m = \alpha$ and $\Gamma^\top \mathbf{1}_n = \beta$ on top of the losses to enforce a balanced matching between objectives and solutions. We could additionally add a regularizer $R(\Gamma)$ to encourage a more diverse assignment. Our many-objective multi-solution transport (MosT) objective is as follows. Find $\{\theta_{1:m}\}$ such that every $\theta_j$ ($j \in [m]$) is the Pareto solution of $m$ weighted objectives

$$\min_{\theta_j} \left( \Gamma_{1,j} L_1(\theta_j), \cdots, \Gamma_{n,j} L_n(\theta_j) \right), \text{ where} \tag{1}$$

$$\Gamma \in \arg\min_{\Gamma \in \Omega} \sum_{i \in [n]} \sum_{j \in [m]} \Gamma_{i,j} L_i(\theta_j) + \tau R(\Gamma), \tag{2}$$

$$\Omega \triangleq \{ \Gamma \in \mathbb{R}_+^{n \times m} : \Gamma \mathbf{1}_m = \alpha, \Gamma^\top \mathbf{1}_n = \beta \}. \tag{3}$$

$\alpha \in \Delta^n$, $\beta \in \Delta^m$ are two tunable vectors on $n$- and $m$-dimensional probability simplexes, respectively. We encourage nondegeneracy of solutions by setting these vectors to follow uniform distributions, i.e., $\alpha = \mathbf{1}_n/n, \beta = \mathbf{1}_m/m$. As discussed in Section 1, the constraint set $\Omega$ prevents the undesired outcome where all objectives are matched with a subset of solutions or all solutions optimize a subset of objectives. To explicitly encourage the diversity among the columns in $\Gamma$, we define a regularization term $R(\cdot)$ as

$$R(\Gamma) = -\sum_{i \in [n]} \max_{j \in [m]} \Gamma_{i,j}. \tag{4}$$

Hence, only the maximum entry $\max_{j\in[m]} \Gamma_{i,j}$ in each row of $\Gamma$ contributes to $R(\Gamma)$. Under the marginal constraints on $\Gamma$ in Eq 3, minimizing $R(\Gamma)$ leads to $\max_{j\in[m]} \Gamma_{i,j} = 1/n$ and zeros for the rest entries in each row-$i$, resulting in zero cosine similarity between different columns of $\Gamma$. Moreover, if $n \gg m$ and $n$ (mod) $m = 0$, it has exactly $n/m$ nonzero entries (with value $1/n$) per column, securing an equal and disjoint partition of the $n$ objectives to the $m$ solutions, which indicates the diversity of objectives used to train the $m$ solutions. Eq. 2 has a weight $\tau$ to balance $R(\Gamma)$ and the transport cost. In the unregularized case when $\tau = 0$ (Eq. 2), the resulting $\Gamma$ would be a sparse matrix (Proposition 1) (Liu et al., 2022a; Brualdi, 2006). Although enforcing marginal constraints *without* $R(\Gamma)$ already leads to improvements over the baselines (Figure 9), empirically, we also showcase that leveraging this extra regularizer further benefits the diverse trade-offs among all objectives (Appendix A.5.3).

## 3.1 Algorithms for MoST

At a high level, the bi-level optimization problem described above can be decoupled into two sub-problems (over $\Gamma$ and $\theta_{1:m}$) when fixing one variable and optimizing the other. At each outer iteration, we first solve 2 exactly by running an off-the-shelf OT solver (e.g., IPOT (Xie et al., 2020)). Then we optimize 1 by running a reweighted version of MGDA with a min-norm solver (Désidéri, 2012). The exact algorithm is summarized in Algorithm 1.

---

**Algorithm 1 M**any-**O**bjective Multi-**S**olution **T**ransport

1: **Input:** objectives $\{L_i(\cdot)\}_{i=1}^n$, $\alpha \in \Delta^n$, $\beta \in \Delta^m$, $\eta$, $K$
2: **Initialize:** $m$ solutions $\theta_{1:m}$
3: **for** $t \in \{1, \cdots, T\}$ **do**
4:     $\Gamma^t \leftarrow$ solution of Eq. 5 by an optimal transport (OT) solver given $\theta_{1:m}^t$;
5:     **for** $j \in \{1, \cdots, m\}$ **do**
6:         **for** $k \in \{1, \cdots, K\}$ **do**
7:             $d_j \leftarrow$ Eq. 8, where $\lambda^*$ is achieved by a min-norm solver for Eq. 7 given $\Gamma^t$ and $\theta_j$.
8:             $\theta_j \leftarrow \theta_j + \eta d_j$ ;
9:         **end for**
10:       $d_j^t \leftarrow d_j; \theta_j^t \leftarrow \theta_j$
11:     **end for**
12: **end for**
13: **Return** $\theta_{1:m}^T$

---

From Algorithm 1, in each iteration, we first optimize $\Gamma^t$ with $\theta_{1:m}^t$ fixed, i.e., finding the optimal transport (or matching) between the $n$ objectives and the $m$ models by solving the following optimal transport problem with existing algorithms (Xie et al., 2020):

$$\min_{\Gamma\in\Omega} \sum_{j\in[m]} \sum_{i\in[n]} \Gamma_{i,j} L_i(\theta_j) + \tau R(\Gamma). \tag{5}$$

Given the maximum entry per row in $\Gamma$, Eq. 5 reduces to a new optimal transport transport problem with an augmented loss $L_i(\cdot) + \tau \nabla R(\Gamma)$, solvable by an off-the-shelf optimal transport solver.

Fixing the optimal $\Gamma$, we then optimize a reweighted version of MGDA across objectives $(\Gamma_{1,j} L_1(\theta_j), \cdots, \Gamma_{n,j} L_n(\theta_j))$ for each solution $\theta_j$, $j \in [m]$. To find Pareto stationary solutions, similar as MGDA, we aim to find the common-descent directions $d_{1:m}$ for $\theta_{1:m}$ to ensure that all objective values do not increase at each iteration. This reduces to solving $m$ MGDA-type MOO problems (more details in Appendix A.1) in parallel, aiming to find $d_j$ for every solution $j \in [m]$

$$\min_{d_j} \max_{i\in[n]} d_j^\top \Gamma_{i,j} \nabla_{\theta_j} L_i(\theta_j) + \tfrac{1}{2}\|d_j\|_2^2, \tag{6}$$

which rescales each objective's gradient $\nabla_{\theta_j} L_i(\theta_j)$ by $\Gamma_{i,j}$. For simplicity, we will use $\nabla L_i(\theta_j)$ to denote $\nabla_{\theta_j} L_i(\theta_j)$ in the remaining of the paper. The dual of Eq. 6 is a min-norm problem over variable $\lambda \in \Delta^n$ as follows:

$$\min_{\lambda\in\Delta^n} \left\| \sum_{i\in[n]} \lambda_i \Gamma_{i,j} \nabla L_i(\theta_j) \right\|^2, \quad \forall j \in [m], \tag{7}$$

which can be solved by existing Frank-Wolfe algorithms (Fujishige, 1980). Given the optimal dual solution $\lambda^*$ from Eq. 7, the primal solution of $d_j$ ($j \in [m]$) to Eq. 6 can be derived by the following convex combination of the $\Gamma$-weighted gradients:

$$d_j = \sum_{i\in[n]} \lambda_i^* \Gamma_{i,j} \nabla L_i(\theta_j). \tag{8}$$

To understand the benefits of Eq. 7, let us consider the vanilla MGDA method: $\min_{\lambda \in \Delta^n} \left\| \sum_{i \in [n]} \lambda_i \nabla L_i(\theta) \right\|^2$, in which $\lambda$ may be biased towards the objective with a small gradient norm (i.e., a well-optimized objective). In MosT, however, OT in Eq. 5 tends to result in a large $\Gamma_{i,j}$ for a small $L_i(\theta_j)$, thus moving small gradient away from the origin in Eq. 7 (i.e., preventing a well-optimized objective from dominating $d_j$).

The MGDA direction $d_j$ guarantees that every objective with non-zero $\Gamma_{i,j}$ will be improved or remain the same after updating $\theta_j$. After obtaining $d_j$, we update the model parameters $\theta_j$ by moving along this direction (Line 8 in Algorithm 1). Optionally, we can also run such gradient descent steps for $K$ steps in practice under the same $\Gamma$. In our convergence analysis (Section 4), we allow for $K > 1$ and assume a full batch setting with $\nabla L_i(\theta_j)$ evaluated on all the local data of problem $i$, for all $j$'s. Empirically, we report our experiment results based on mini-batch gradients in Section 6.

### 3.2 EXTENSION TO FEW-OBJECTIVE ($n < m$) CASES

The MosT formulation discussed before is mainly motivated by the challenges of having many objectives in MOO. When $n \gg m$, the diversity of the $m$ models can be achieved by enforcing the two marginal constraints in the optimal transport problem. However, the diversity cannot be fully guaranteed when $n \ll m$. For example, when $n = 2$, by even applying uniform distributions for $\alpha$ and $\beta$ (i.e., the strongest constraints for diversity), a trivial but feasible solution of $\Gamma = [\mathbf{1}_{m/2}, \mathbf{0}_{m/2}; \mathbf{0}_{m/2}, \mathbf{1}_{m/2}]$ can collapse the $m$ models to duplicates of only two different models, i.e., one minimizing the first objective while the other minimizing the second. To address this problem, we create $(n' - n) \gg m$ dense interpolations of the $n$ objectives ($n' \gg m$) by sampling $(n' - n)$ groups of convex combination weights $w_{n+1:n'}$ on the simplex, i.e., $w_i \in \Delta^n$ drawn from a Dirichlet distribution. Then each auxiliary objective $L_i(\cdot)$ can be defined as

$$L_i(\theta) \triangleq \sum_{l \in [n]} w_{i,l} L_l(\theta), \ \ \forall \, i = n + 1, \cdots, n'. \tag{9}$$

Thereby, we increase the number of objectives to $n' \gg m$ and MosT can be applied to achieving diverse models for optimizing the $n'$ interpolations of the original $n$ objectives. This strategy can be explained as maximizing the coverage of the $m$ models over the dense samples of the Pareto front regions using $n'$ random reference vectors. We report the results of this technique with applications on a toy problem and fairness/utility trade-offs in Appendix A.3.

### 3.3 A PRACTICAL SOLUTION-SPECIALIZATION CURRICULUM

In scenarios with diverse objectives, each corresponding to a distinct domain or unique dataset, a practical demand arises: optimizing multiple models and turning them into a mixture of specialized experts (i.e., models). This allows each input sample to select the best expert(s) for inference. Such "objective selecting expert/models" or "objective choice routing" strategy corresponds to removing $\Gamma \mathbf{1}_n = \beta$ from the constraint set $\Omega$ in Eq. 3. But it may lead to training imbalance among the $m$ models, e.g., one model is chosen by most objectives while other models get nearly zero optimization. As training proceeds, the winning model(s) trained by more objectives tend to be chosen even more frequently; hence joint optimization of $m$ models can collapse to training one single model.

To address this challenge, we propose a curriculum of varying the marginal constraints that progressively changes $\alpha$ and $\beta$ for different training stages. Initially, we mainly focus on enforcing a uniform marginal distribution $\beta$ so every model receives sufficient training from multiple objectives. By relaxing the other marginal constraint $\alpha$ over $n$ objectives to be slightly non-uniform, the $m$ models have more freedom to choose the objectives they perform the best on (i.e., "model selecting objective" or "model choice routing") and we allow for slight imbalance among objectives.[2] In later stages, the curriculum instead focuses more on enforcing a uniform $\alpha$ so every objective is covered by sufficient models. The marginal constraint $\beta$ can be relaxed in this stage since models approach convergence. Empirical results in Appendix A.5.2 demonstrate the effectiveness of our curriculum strategy.

## 4 PROPERTIES OF MOST

In this section, we discuss how MosT encourages diverse solutions and provide convergence guarantees of Algorithm 1. Here we define solution diversity via the diversity of $\Gamma$, as stated below.

---

[2]Less-selected objectives can be more difficult and it is preferable to learn them later when models improve.

**Definition 1** (Diverse Solutions). We informally say that a set of solutions $\{\theta_i\}_{i \in [n]}$ are more diverse if $\sum_{j=1}^{m} \sum_{z=1, z \neq j}^{m} cos(\Gamma_{\cdot,j}, \Gamma_{\cdot,z})$ is small with some feasible $\Gamma$, where $cos(\boldsymbol{x}, \boldsymbol{y}) = \frac{\boldsymbol{x} \cdot \boldsymbol{y}}{||\boldsymbol{x}|| \cdot ||\boldsymbol{y}||}$.

This definition captures the goal of diversifying solution specialization, ensuring at least one suitable model $j$ (corresponding to $\arg\max_{j \in [m]} \Gamma_{i,j}$) for every objective $i$, as losses are reweighted by $\Gamma$. We note that fixing the losses $\{L_i(\theta_j)\}_{i \in [n], j \in [m]}$, when solving for $\Gamma$ without any regularization (i.e., $\tau{=}0$ in Eq. 5), the objective inherently results in sparse solutions, as stated in the proposition below.

**Proposition 1** (Sparsity of $\Gamma$ Brualdi (2006)). Any $\Gamma$ that solves the optimal transport problem Eq. 5 with $\tau{=}0$ has at most $n + m - 1$ non-zero entries.

Intuitively, sparse $\Gamma$'s satisfying the marginal constraint $\Omega$ would prevent the scenarios where only a subset of objectives are well-optimized. In Appendix A.5.3, we empirically verify that even without $R(\Gamma)$, the sparse transport gives diverse solutions that balance all objectives. Also, setting $R(\Gamma)$ to be the negative of our diversity measure (Definition 1), further encourage diversity as it is obvious to show that

$$\sum_{i \in [n]} \max_{j \in [m]} \Gamma_{i,j}^*(\tau) \geq \sum_{i \in [n]} \max_{j \in [m]} \Gamma_{i,j}^*(0),$$

where $\Gamma^*(\tau)$ is defined as the optimal solution of Eq. 5 with a regularization constant $\tau$. We investigate training dynamics and diverse assignments between objectives and models in Section 5.

### 4.1 CONVERGENCE

In this part, we analyze the convergence of MosT in Algorithm 1 for both strongly-convex and non-convex functions. The alternate minimization scheme poses additional challenges to our analysis compared with prior convergence results in MOO (Fliege et al., 2019).

**Theorem 1** (Convex and Non-Convex). Assume each objective $L_i(\theta)$ is $\nu$-smooth. Given marginal distribution constraints $\alpha \in \Delta^n$ and $\beta \in \Delta^m$, under a learning rate $\eta = \frac{1}{2\nu}$, after running Algorithm 1 for $T$ outer iterations with full batch multi-gradient descent, we have that

$$\frac{1}{T} \sum_{t \in [T]} \sum_{j \in [m]} \beta_j \|d_j^t\|^2 \leq O\left(\frac{\nu}{T}\right).$$

We defer the full proofs to Appendix A.2.2. The main step involves leveraging the property of the common descent direction $d_j$ obtained from Eq. 8, i.e., for any $i \in [n]$ and $j \in [m]$, we have

$$\Gamma_{i,j}^t \nabla L_i(\theta_j^t)^\top d_j^t \leq -\frac{1}{2} \|d_j^t\|^2.$$

Our convergence rate is the same as that of normal gradient descent for non-convex and smooth problems under a fixed learning rate. Note that our result uses full gradients of each objective for the min-norm problem (Eq. 6). In practice, we use $K > 1$ to run multiple iterations to update the model parameters for each objective locally, and use stochastic mini-batch gradients in Eq. 6.

**Strongly-convex cases.** When the losses $L_i(\theta)$ ($i \in [n]$) is smooth and additionally strongly convex with respect to $\theta \in \mathbb{R}^d$, we show that our proposed algorithm (using full gradients) can converge to Pareto stationary points with respect to a subset of matching objectives under a more restricted assumption on the stability of objective-model matching. Please see Theorem 2 in the appendix for complete statement and a detailed proof.

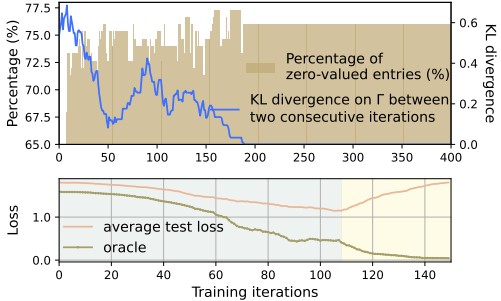

Figure 2: (a) Left y-axis: Percent of zero-valued entries within $\Gamma$. Right y-axis: symmetric KL divergence between $\Gamma$ in successive iterations. $\Gamma$ **quickly converges to sparsity**. (b) Test loss averaged over all solutions vs. test loss of the best-performing solution for each objective (oracle). As training proceeds, average loss rises while oracle loss continues to decrease, indicating a trend of **solution specialization and diversification**.

### 5 ASSIGNMENT DYNAMICS DURING TRAINING

In this section, we empirically examine the optimal assignment matrix $\Gamma$ during training, highlighting its advantages on objective-solution matching and its impact on solution specialization.

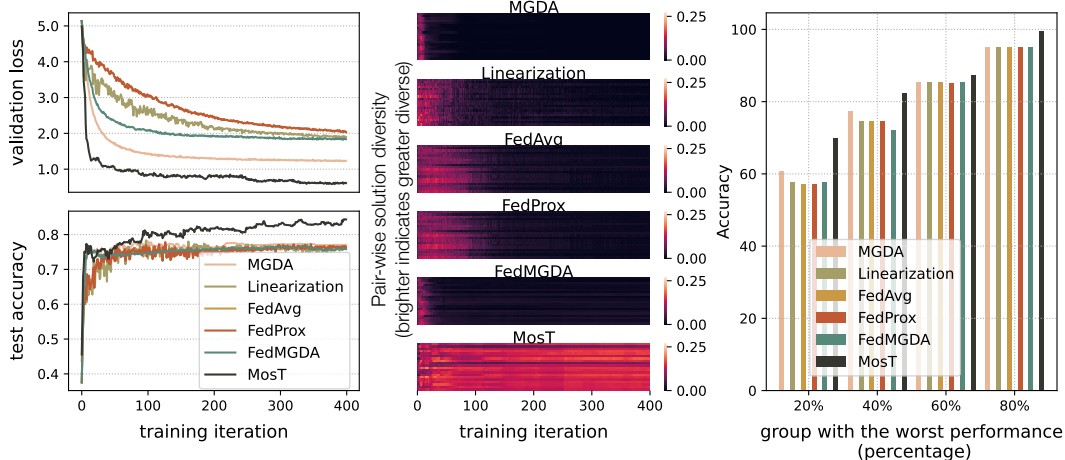

Figure 3: (a) **Training loss and test accuracy** curves of each method. MosT demonstrates faster convergence with higher accuracy. (b) **Diversity** of solutions during training: each block on a column visualizes the KL divergence of a pair of solutions (brighter indicates a larger value). MosT produces more diverse solutions. (c) **Fairness:** Accuracy of the worst 20%, 40%, 60%, and 80% client groups. Diversity leads to better tail performance among all the objectives.

**Fast convergence of $\Gamma$.** We observe rapid convergence of $\Gamma$ by tracking its evolution with Kullback Leibler (KL) divergence between successive iterations (Figure 2(a)), detailed in Appendix A.4.

**Sparsity of $\Gamma$.** As training proceeds, $\Gamma$ becomes more sparse, with nearly 75.00% zero entries (Figure 2(a)). This implies that $\Gamma$ prompts each solution to focus on only a subset of objectives.

**Sparsity promotes specialization.** Figure 2(b) shows the mean test loss averaged across all solutions evaluated at all the objectives, and the loss when selecting the best-performing solution for each objective (denoted as *oracle*). Initially, the mean loss increases and then decreases, while the oracle loss consistently decreases. These trends indicate that using all the solutions to serve all the objectives is suboptimal, while objective-specific solutions can focus on subsets of objectives, ultimately contributing to a more complementary and effective overall solution set.

## 6 MOST APPLICATIONS

In this section, we apply MosT to various ML applications where $n \gg m$. Though we do not specifically focus on the (less challenging) settings of $m \gg n$, MosT can naturally generate diverse solutions for those problems as well. We defer the readers to Appendix A.3, where we show superior performance of MosT relative to existing methods for the $m \gg n$ case.

### 6.1 EXPERIMENTAL SETUP

This section outlines the experimental setup. We describe the specific setup for each application in their respective sections, and provide more details such as hyperparameter values in Appendix A.4.

For all the applications, we *at least* compare with the following baselines to generate $m$ solutions.

- **Linearization-based MOO** (abbr. *Linearization* below) where we optimize a convex combination of all objectives with $m$ randomly-sampled sets of weights. Minimizing a simple average of all the losses (empirical risk minimization over objectives) is a specific instance of using uniform weights.
- Running **MDGA** (Désidéri, 2012) independently for $m$ times with different random seeds.

**Evaluation metrics.** We use task-specific evaluation metrics, such as average accuracy (or tail accuracy) across all objectives or hypervolume (Zitzler & Thiele, 1999). We do not evaluate on hypevolume for many-objective scenarios due to its computational complexity and infeasibility in high-dimensional settings (i.e., many objectives). Each run is repeated three times using different seeds.

Table 1: **Federated Learning:** Mean accuracy across clients (mean and std across 3 runs) on federated learning datasets. MosT outperforms the strong baselines.

| Dataset | MGDA | Linearization | FedAvg | FedProx | FedMGDA+ | MosT w/o $R(\Gamma)$ | MosT |
|---------|------|---------------|--------|---------|----------|----------------------|------|
| Syn (0.0, 0.0) | $77.22_{\pm0.41}$ | $75.91_{\pm0.37}$ | $75.71_{\pm0.51}$ | $75.60_{\pm0.42}$ | $75.26_{\pm1.21}$ | $83.09_{\pm0.87}$ | $\mathbf{84.25}_{\pm0.51}$ |
| Syn (0.5, 0.5) | $87.09_{\pm0.29}$ | $87.18_{\pm0.27}$ | $86.26_{\pm0.61}$ | $86.13_{\pm0.39}$ | $85.21_{\pm1.42}$ | $89.07_{\pm0.63}$ | $\mathbf{89.99}_{\pm0.52}$ |
| Syn (1.0, 1.0) | $90.52_{\pm0.13}$ | $89.87_{\pm0.51}$ | $88.12_{\pm0.75}$ | $87.58_{\pm1.36}$ | $87.16_{\pm1.09}$ | $91.70_{\pm0.02}$ | $\mathbf{92.21}_{\pm0.08}$ |
| FEMNIST | $78.86_{\pm1.43}$ | $72.62_{\pm0.65}$ | $72.47_{\pm0.19}$ | $72.45_{\pm0.06}$ | $80.08_{\pm0.12}$ | $80.94_{\pm0.34}$ | $\mathbf{81.16}_{\pm0.03}$ |

## 6.2 FEDERATED LEARNING

One important scenario where $n \gg m$ is the cross-device federated learning application, where we jointly learn $m$ models over a heterogeneous network of $n$ remote devices. The devices generate local data following non-identical distributions; hence we view the finite sum of empirical losses on each device as one objective, i.e., $L_i(\theta) := \frac{1}{v_i} \sum_{s=1}^{v_i} l_s(\theta)$ where $v_i$ is the number of local samples on device $i \in [n]$, and $l_s$ denotes the individual local loss on sample $s$. MosT seamlessly integrates with the decentralized setting of federated learning by computing client-specific local updates $(\theta_{1:m})$ and aggregating them to update the global model by $\Gamma$. Moreover, since clients are diverse, it is expected that the solution diversity benefits of MosT will contribute significantly to the final performance.

We conduct experiments on synthetic data and Federated Extended MNIST (FEMNIST) (Cohen et al., 2017; Caldas et al., 2018), where the number of objectives $n = 30$ and $n = 206$, respectively. We experiment on three synthetic datasets, denoted as Syn $(\rho_1, \rho_2)$, with different $\rho_1$ and $\rho_2$ controlling heterogeneity of local models and data, as detailed in Appendix A.4. We compare MosT with baselines described in Section 6.1 and state-of-the-art federated learning algorithms, including FedAvg (McMahan et al., 2017), FedProx (Li et al., 2020), and FedMGDA+ (Hu et al., 2022). We run each algorithm $m$ times with different random initializations. It is worth noting that during evaluation, for all methods, we let each device pick a best model out of $m$ solutions based on the validation set, and compute its performance. We then report the average and the quantile accuracy across all devices. As the results shown in Table 1, MosT outperforms the baselines by a large margin on all datasets. Furthermore, we have the following observations.

**MosT results in significant convergence improvements.** Figure 3(a) compares the training loss and test accuracy curves of different algorithms. Notably, MosT outperforms baselines with a lower training loss (faster convergence) and higher test accuracy (better generalization to unseen data).

**MosT maintains diversity during training.** We analyze how the diversity of solutions evolves for different algorithms. Diversity is quantified using the KL divergence between predictions of any pair of solutions generated by each algorithm. Figure 3(b) shows that initially, all algorithms exhibit high diversity due to randomized initialization. However, baselines witness a notable decrease in solution diversity during training. In contrast, MosT maintains high diversity throughout the training process.

**MosT promotes fairness in FL.** MosT's improved diversity is anticipated to benefit clients typically overlooked by other algorithms, thus enhancing fairness. To validate, we calculate the accuracy of the worst 20%, 40%, 60%, and 80% clients for each algorithm. As depicted in Figure 3(c), MosT outperforms the baselines by a larger margin for clients with worse performance, which demonstrates that the diversity of MosT effectively promotes fairness in FL.

Furthermore, our study reveals that MosT strategically assigns diverse solutions for inference, preventing the collapse phenomenon in MOO (detailed in Appendix A.5.1).

## 6.3 MULTI-TASK LEARNING

MosT seamlessly extends its capabilities to multi-task learning by treating each task as an individual objective. Our experiments explore two real-world datasets, Office-Caltech10 (Saenko et al., 2010; Griffin et al., 2007) and DomainNet (Peng et al., 2019) with $n = 4$ and 6 objectives, respectively. We have $m = 3$ solutions for Office-Caltech10 and $m = 4$ for DomainNet. We conduct a thorough comparison with various state-of-the-art multi-task learning approaches: MGDA, Linearization-based MOO, EPO which is based on user preference vectors (Mahapatra & Rajan, 2020), COSMOS (Ruchte & Grabocka, 2021), and TAG which identifies task grouping for MOO (Fifty et al., 2021). Similar to federated learning datasets, we select the best-performing solutions over the validation set for

inference and report the accuracy averaged across all tasks. The results are shown in Table 2, which demonstrates MosT's superiority over existing approaches.

Table 2: **Multi-task Learning:** Average accuracy across all tasks (mean and std across 3 runs) on Office-Caltech10 and DomainNet.

| Dataset | MGDA | Linearization | EPO | COSMOS | TAG | MosT |
|---|---|---|---|---|---|---|
| Office-10 | $80.74_{\pm 0.44}$ | $61.26_{\pm 0.67}$ | $61.05_{\pm 1.09}$ | $63.83_{\pm 1.01}$ | $49.38_{\pm 1.10}$ | $\mathbf{82.98}_{\pm 0.51}$ |
| DomainNet | $65.81_{\pm 0.37}$ | $57.15_{\pm 0.17}$ | $58.55_{\pm 0.37}$ | $63.78_{\pm 0.34}$ | $31.05_{\pm 1.24}$ | $\mathbf{67.65}_{\pm 0.55}$ |

## 6.4 MIXTURE-OF-PROMPT LEARNING

Another application is prompt learning for language models (Qin & Eisner, 2021), where solutions need to generalize well with diverse instances. To address this, MosT trains $m$ soft prompts to handle each training instance as a distinct objective, with $n$ being the total number of instances. We define the objective function as $L_i(\theta) := l_i(\theta)$, where $l_i$ represents the loss on instance $i$. This allows MosT to tailor its learning process to each specific sample, adapting to diverse linguistic nuances. We experiment on three datasets from the SuperGLUE benchmark (Wang et al., 2019). For all datasets, we sample $n = 128$ training instances, while generating $m = 3$ soft prompts.

Note that MosT prioritizes training complementary solutions for multiple objectives, rather than their assignment. Simple ways, like selecting the best-performing solution over the validation set, are promising for scenarios like federated learning and multi-task learning. However, treating each instance as an objective poses challenges in solution selection during inference. To address this, we train a dispatcher for each algorithm to learn correlations between prompts and instances (implementation details in Appendix A.4), selecting the highest correlated one for inference. We report average accuracy across all tasks, in Table 3. Results show MosT exhibits a significantly better ability to generalize to unseen instances compared to baselines.

## 6.5 ABLATION STUDIES AND COMPARISON WITH OTHER BASELINES

We conduct extensive ablation studies to validate the design of MosT, examining the necessity of OT (Appendix A.5.1) and MGDA (Appendix A.5.4), along with specific designs like solution-specialization curriculum (Appendix A.5.2) and diversity encouragement (Appendix A.5.3). These studies offer insights into the effectiveness of components of MosT:

**Necessity of OT and MGDA.** Comparing OT-generated and randomly generated weights reveals the necessity of OT for achieving balanced objective-solution matching. Additionally, MGDA consistently outperforms its linearization-based alternative, highlighting its effectiveness in parameter updates for weighted multi-objective optimization.

**Effectiveness of curriculum.** Our proposed scheduling of the marginal constraints (Section 3.3) boosts performance by over 2.00% and enhances training stability.

**Benefits of diversity-encouragement regularization.** It enhances performance and fairness.

**Preventing collapse with OT.** Comparison of three strategies introduced in Section 3.3, MosT, "objective selecting model", and "model selecting objective", reveals the collapse phenomenon in MOO, where limited solutions dominate all objectives. In contrast, MosT using OT with a two-way matching approach achieves a more balanced distribution of objectives among models during training.

## 6.6 RUNTIME COMPARISONS

We enhance the computational efficiency of our algorithm employing several simple practices: 1) detecting early convergence of OT, 2) initializing transport plans using prior computations, 3) running MGDA on a subset of model parameters like the final layer's bias term in a neural network, and 4) effectively reducing objectives in MGDA through the inherent sparsity of regularized OT.

Table 4 presents the end-to-end runtime (in seconds) of different approaches on federated learning datasets with $n = 30$ and $n = 206$. We see that the running time of MosT is comparable to that of baseline methods. Notably, the time needed for OT across datasets is negligible due to the use of

Table 3: **Mixture-of-Prompt Learning:** Test accuracy on three datasets of SuperGLUE benchmark (mean and std across 3 runs).

| Task | MGDA | Linearization | MosT |
|------|------|---------------|------|
| BoolQ | $62.69_{\pm 0.71}$ | $61.30_{\pm 0.09}$ | $\mathbf{67.03}_{\pm 0.49}$ |
| MultiRC | $60.86_{\pm 0.50}$ | $58.79_{\pm 1.23}$ | $\mathbf{63.78}_{\pm 0.15}$ |
| WiC | $55.28_{\pm 0.89}$ | $57.16_{\pm 1.27}$ | $\mathbf{62.38}_{\pm 0.31}$ |

existing packages and the fast convergence of $\Gamma$ (shown in Section 5), accounting for less than 1% of the total time. See Appendix A.6 for a complete analysis of computation time in other applications.

Table 4: **Runtime** comparison (sec) on federated learning datasets.

| Dataset | MGDA | Linearization | FedAvg | FedMGDA+ | MosT |
|---------|------|---------------|--------|----------|------|
| Syn (0.0) | 219.86 | 225.90 | 222.22 | 516.25 | 217.59 |
| Syn (0.5) | 208.82 | 208.27 | 205.90 | 495.62 | 201.70 |
| Syn (1.0) | 269.44 | 268.33 | 270.65 | 557.58 | 260.50 |
| FEMNIST | 3522.83 | 3135.76 | 3147.62 | > 5000.00 | 3368.63 |

## 7 CONCLUSION

In this paper, we propose "many-objective multi-solution transport (MosT)", aiming to find $m$ Pareto solutions (models) that achieve diverse trade-offs among $n$ optimization objectives. We investigate a challenging case of $n \gg m$, where existing methods often struggle with exploring a high-dimensional Pareto frontier. We formulate MosT as a bi-level optimization of multiple weighted objectives, where weights guide the exploration and are determined by an optimal transport (OT) matching objectives and solutions. Our algorithm theoretically converges to $m$ Pareto solutions by alternating between optimizing the weighted objectives and OT. MosT extends to achieve diverse solutions for $n \ll m$. We apply MosT to various machine learning problems, training $m$ models to serve $n$ users, domains, or criteria. Empirically, MosT outperforms other strong baselines in tasks like federated learning, multi-task learning, mixture-of-prompt, fairness-accuracy trade-offs, and other MOO benchmarks.

## ACKNOWLEDGMENTS

We would like to thank Shengjie Wang, Xin Yang, and Yuanyuan Yang for their insightful discussions in the earlier-stage exploration of this project's initial idea. We appreciate the reviewers and area chairs for their constructive comments and suggestions.

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

# A  APPENDIX

## A.1  BACKGROUND ON MGDA IN MULTI-OBJECTIVE OPTIMIZATION

We first describe some background on the multi-gradient descent algorithm to solve multi-objective optimization.

Let $\mathbf{L}(\theta) \in \mathbb{R}^n$ be defined as

$$\mathbf{L}(\theta) := (L_1(\theta), \cdots, L_n(\theta)), \theta \in \mathbb{R}^d. \tag{10}$$

The goal for the multi-objective optimization (minimization) problem is to find Pareto optimal solutions with respect to all objectives $L_i(\cdot), i \in [n]$. One line of method is at each iteration, to find a common descent direction $d$ for all objectives. Given the current model $\theta$, we would like to find a descent step to minimize each objective value. For the single-objective case, the direction is $-\nabla L(\theta)$. For $n$ objectives, one objective is to solve for $d$:

$$\min_d \left\{ \max_{i \in [n]} \nabla L_i(\theta)^\top d + \frac{1}{2} \|d\|^2 \right\}, \tag{11}$$

and then apply $d$ as $\theta = \theta + \eta d$. If the optimal objective value of Eq. 11 is negative, then there exists a descent direction $d^*$ such that all objective values will be decreased. If $\theta$ is Pareto stationary, then $d = \mathbf{0}$ and the optimal objective value is 0. This formulation is equivalent to

$$\min_{b,d} \quad b + \frac{1}{2} \|d\|^2 \tag{12}$$
$$s.t. \quad \nabla L_i(\theta)^\top d \leq b, i \in [n].$$

Formally, we have the following lemma.

**Lemma 1** (Good Descent Direction Désidéri (2012) ). Let $d, b$ be the solutions of Eq. 12, then

1. If $\theta$ is Pareto stationary, then $d = \mathbf{0}$ and $b = 0$.

2. If $\theta$ is not Pareto stationary, then

$$b \leq -\frac{1}{2} \|d\|^2 < 0, \tag{13}$$
$$\nabla L_i(\theta)^\top d \leq b, \ i \in [n]. \tag{14}$$

**Lemma 2** (A Rescaled Version of Lemma 1). Let $d_j \in \mathbb{R}^d$ be the solution of Eq. 6 and $\Gamma_{i,j}$ be some non-negative scalar, then

1. If $\theta_j$ is Pareto stationary, then $d_j = \mathbf{0}$.

2. If $\theta_j$ is not Pareto stationary, then

$$\Gamma_{i,j} L_i(\theta_j)^\top d_j \leq -\frac{1}{2} \|d_j\|^2, \ i \in [n]. \tag{15}$$

## A.2  CONVERGENCE PROOFS

### A.2.1  STRONGLY-CONVEX CASES

MosT learns a matching between objectives and solutions represented by its non-zero entries. Therefore, we show that our proposed algorithm (using full gradients) can converge to Pareto stationary points with respect to a subset of matching objectives in strongly-convex cases. We first make a common assumption.

**Assumption 1.** Each objective $L_i(\theta)$ ($i \in [n]$) is $\nu$-smooth and $\mu$-strongly convex w.r.t. $\theta \in \mathbb{R}^d$.

We introduce another assumption on objective-model matching, as follows.

**Assumption 2.** There exists an $s$ ($s < \infty$) such that after $s$ iterations, all the non-zero entries in $\Gamma^s$ remain non-zero, which are lower bounded by a small constant $\epsilon$.

This assumption can be interpreted as that the assignment of solutions for each objective become stable during training. From Figure 2, we empirically observe that after certain iterations, the learnt $\Gamma$ will have the same non-zero patterns.

**Theorem 2** (Strongly-Convex). Let Assumption 1 and 2 hold. Given marginal distribution constraints $\alpha \in \Delta^n$ and $\beta \in \Delta^m$, under a fixed learning rate $\eta \leq \frac{1}{\nu}$, after running Algorithm 1 for $T + s$ outer iterations with full multi-gradient descent, we have for each solution $j \in [m]$,

$$\left\| \theta_j^{T+s} - \theta_j^* \right\|^2 \leq (1 - \mu\eta\epsilon)^{TK} \left\| \theta_j^s - \theta_j^* \right\|^2, \tag{16}$$

where $\theta_j^*$ is a Pareto stationary solution across objectives with a non-zero $\Gamma_{\cdot,j}^s$.

*Proof.* First, let us assume $K = 1$. At each iteration $t$, we have

$$d_j^t = -\sum_{i \in [n]} \lambda_i \Gamma_{i,j}^t \nabla L_i(\theta_j^t), \quad \forall j \in [m] \tag{17}$$

for some $\{\lambda_i\}_{i \in [n]} \in \Delta^n$ which is the solution of Eq. 7. First we note that for every $j \in [m]$, $\theta_j$ will converge. If $d_j^t = \mathbf{0}$, then $\theta_j$ has converged to a Pareto stationary point. Otherwise, for every objective $L_i$, we have monotonically decrease for the sequence: $L_i(\theta_j^{t+1}) - L_i(\theta_j^t) \leq -\frac{1}{2}\eta(1 - \nu\eta)\|d_j^t\|^2 < 0$. Therefore, every solution will converge. We denote $\theta_j^*$ as one of the Pareto stationary solutions of Eq. 1 that solution $\theta_j$ converges to. By definition and the properties of MDGA, we know that for every solution, every objective value will be non-increasing throughout optimization. Hence, for every $i \in [n]$ and $j \in [m]$, it holds that

$$L_i(\theta_j^{t+1}) - L_i(\theta_j^*) \geq 0. \tag{18}$$

For every $i \in [n]$, the $\nu$-smoothness and $\mu$-convexity of $L_i$ lead to

$$L_i(\theta_j^{t+1}) = L_i(\theta_j^t + \eta d_j^t) \tag{19}$$

$$\leq L_i(\theta_j^t) + \eta \nabla L_i(\theta_j^t)^\top d_j^t + \frac{\nu}{2}\|\eta d_j^t\|^2 \tag{20}$$

$$\leq L_i(\theta_j^*) + \nabla L_i(\theta_j^t)^\top (\theta_j^t - \theta_j^*) - \frac{\mu}{2}\left\| \theta_j^t - \theta_j^* \right\|^2 + \eta \nabla L_i(\theta_j^t)^\top d_j^t + \frac{\nu}{2}\|\eta d_j^t\|^2. \tag{21}$$

By moving $L_i(\theta_j^*)$ to the left-hand side and multiplying both sides by $\lambda_i \Gamma_{i,j}^t$, we have

$$\sum_{i \in [n]} \lambda_i \Gamma_{i,j}^t \left( L_i(\theta_j^{t+1}) - L_i(\theta_j^*) \right) \tag{22}$$

$$\leq \sum_{i \in [n]} \lambda_i \Gamma_{i,j}^t \nabla L_i(\theta_j^t)^\top (\theta_j^t - \theta_j^* + \eta d_j^t) - \sum_{i \in [n]} \lambda_i \Gamma_{i,j}^t \frac{\mu}{2} \left\| \theta_j^t - \theta_j^* \right\|^2 + \sum_{i \in [n]} \lambda_i \Gamma_{i,j}^t \frac{\nu}{2}\|\eta d_j^t\|^2. \tag{23}$$

As $\Gamma_{i,j}^t \geq \epsilon > 0$, then

$$\sum_{i \in [n]} \lambda_i \Gamma_{i,j}^t \frac{\mu}{2}\|\theta_j^t - \theta_j^*\|^2 \leq \frac{\mu\epsilon}{2}\|\theta_j^t - \theta_j^*\|^2. \tag{24}$$

Due to the Hölder inequality, we have $\sum_{i \in [n]} \lambda_i \Gamma_{i,j} \le \|\lambda\|_1 \|\Gamma_{\cdot,j}\|_\infty := \beta_j \le 1$. Hence, we have

$$
\sum_{i \in [n]} \lambda_i \Gamma_{i,j}^t \left( L_i(\theta_j^{t+1}) - L_i(\theta_j^*) \right)
$$

$$
\le \sum_{i \in [n]} \lambda_i \Gamma_{i,j}^t \nabla L_i(\theta_j^t)^\top (\theta_j^t - \theta_j^* + \eta d_j^t) - \frac{\mu \epsilon}{2} \left\| \theta_j^t - \theta_j^* \right\|^2 + \frac{\nu \beta_j}{2} \| \eta d_j^t \|^2 \tag{25}
$$

$$
= -d_j^t (\theta_j^t - \theta_j^*) - \eta \| d_j^t \|^2 - \frac{\mu \epsilon}{2} \left\| \theta_j^t - \theta_j^* \right\|^2 + \frac{\nu \beta_j}{2} \| \eta d_j^t \|^2
$$

$$
\le -d^t (\theta_j^t - \theta_j^*) - \eta \left( 1 - \frac{\beta_j}{2} \right) \| d_j^t \|^2 - \frac{\mu \epsilon}{2} \left\| \theta_j^t - \theta_j^* \right\|^2 \text{ (taking } \eta \le \frac{1}{\nu}) \tag{26}
$$

$$
\le -d^t (\theta_j^t - \theta_j^*) - \frac{\eta}{2} \| d_j^t \|^2 - \frac{\mu \epsilon}{2} \left\| \theta_j^t - \theta_j^* \right\|^2 \text{ (using } \beta_j \le 1)
$$

$$
= -\frac{1}{2\eta} (2\eta d_j^t (\theta_j^t - \theta_j^*) + \| \eta d_j^t \|^2) - \frac{\mu \epsilon}{2} \left\| \theta_j^t - \theta_j^* \right\|^2
$$

$$
= -\frac{1}{2\eta} (2(\theta_j^{t+1} - \theta_j^t)^\top (\theta_j^t - \theta_j^*) + \| \theta_j^{t+1} - \theta_j^t \|^2) - \frac{\mu \epsilon}{2} \left\| \theta_j^t - \theta_j^* \right\|^2
$$

$$
= -\frac{1}{2\eta} (2\theta_j^{t+1} \theta_j^t - 2\| \theta_j^t \|^2 - 2(\theta_j^{t+1} - \theta_j^t)^\top \theta_j^* + \| \theta_j^{t+1} \|^2 + \| \theta_j^t \|^2 - 2\theta_j^{t+1} \theta_j^t) - \frac{\mu \epsilon}{2} \left\| \theta_j^t - \theta_j^* \right\|^2
$$

$$
= -\frac{1}{2\eta} (\| \theta_j^{t+1} \|^2 - 2(\theta_j^{t+1} - \theta_j^t)^\top \theta_j^* - \| \theta_j^t \|^2) - \frac{\mu \epsilon}{2} \left\| \theta_j^t - \theta_j^* \right\|^2
$$

$$
= -\frac{1}{2\eta} (\| \theta_j^{t+1} - \theta_j^* \|^2 - \| \theta_j^t - \theta_j^* \|^2) - \frac{\mu \epsilon}{2} \left\| \theta_j^t - \theta_j^* \right\|^2
$$

$$
= \frac{1}{2\eta} (\| \theta_j^t - \theta_j^* \|^2 - \| \theta_j^{t+1} - \theta_j^* \|^2) - \frac{\mu \epsilon}{2} \left\| \theta_j^t - \theta_j^* \right\|^2. \tag{27}
$$

Since during optimization, we guarantee that every objective value will be non-increasing at each iteration, we have $L_i(\theta_j^{t+1}) - L_i(\theta_j^*) \ge 0$. So the left-hand side of Eq. 22 is non-negative. Hence,

$$
\frac{1}{2\eta} (\| \theta_j^t - \theta_j^* \|^2 - \| \theta_j^{t+1} - \theta_j^* \|^2) - \frac{\mu \epsilon}{2} \left\| \theta_j^t - \theta_j^* \right\|^2 \ge 0, \tag{28}
$$

$$
\| \theta_j^{t+1} - \theta_j^* \|^2 \le (1 - \mu \eta \epsilon) \| \theta_j^t - \theta_j^* \|^2, \tag{29}
$$

which gives us linear convergence.

When $K > 1$, at each outer iteration, fixing $\Gamma_{i,j}^t$, we are running multiple updates on the model parameters. In this case, we still have

$$
\| \theta_j^{t,k+1} - \theta_j^* \| \le (1 - \mu \eta \epsilon) \| \theta_j^{t,k} - \theta_j^* \|^2, \tag{30}
$$

where $\| \theta_j^{t,k+1} \|$ denote the model parameters at the $t$-th outer iteration and $k+1$-th inner iteration (Line 6 of Algorithm 1). Hence,

$$
\| \theta_j^{t+1} - \theta_j^* \| \le (1 - \mu \eta \epsilon)^K \| \theta_j^t - \theta_j^* \|^2 \tag{31}
$$

holds. □

### A.2.2 Non-Convex and Smooth Cases

For simplicity, we first consider the case where $K = 1$. From Lemma 2, we know that at each iteration $t$,

$$
\Gamma_{i,j}^t \nabla L_i(\theta_j^t)^\top d_j^t \le -\frac{1}{2} \left\| d_j^t \right\|^2, \quad \forall i \in [n]. \tag{32}
$$

Assuming $\nu$-smooth of each $L_i$, we have

$$\Gamma_{i,j}^t \left( L_i(\theta_j^{t+1}) - L_i(\theta_j^t) \right) = \Gamma_{i,j}^t \left( L_i(\theta_j^t + \eta d_j^t) - L_i(\theta_j^t) \right) \tag{33}$$

$$\leq \eta \Gamma_{i,j}^t \nabla L_i(\theta_j^t)^\top d_j^t + \frac{\nu \Gamma_{i,j}^t}{2} \|\eta d_j^t\|^2 \tag{34}$$

$$\leq -\frac{\eta}{2} \|d_j^t\|^2 + \frac{\nu \eta^2}{2} \Gamma_{i,j}^t \|d_j^t\|^2 \tag{35}$$

$$\leq -\frac{\eta}{2} \Gamma_{i,j}^t \|d_j^t\|^2 + \frac{\nu \eta^2}{2} \Gamma_{i,j}^t \|d_j^t\|^2 \quad (\Gamma_{i,j}^t \leq 1) \tag{36}$$

$$= -\frac{\eta(1 - \nu\eta)}{2} \Gamma_{i,j}^t \|d_j^t\|^2. \tag{37}$$

Sum over all models $j \in [m]$,

$$\sum_{j \in [m]} \Gamma_{i,j}^t \left( L_i(\theta_j^{t+1}) - L_i(\theta_j^t) \right) \leq -\frac{\eta(1 - \nu\eta)}{2} \sum_{j \in [m]} \Gamma_{i,j}^t \|d_j^t\|^2. \tag{38}$$

The above result is for a single objective $L_i(\cdot)$. Now let's consider the weighted sum of all the $n$ objectives between two steps with different $\Gamma$'s, i.e., $\Gamma^t$ and $\Gamma^{t+1}$. By the optimality of $\Gamma^{t+1}$,

$$\sum_{i \in [n]} \sum_{j \in [m]} \Gamma_{i,j}^{t+1} L_j(\theta_j^{t+1}) - \tau \sum_{i \in [n]} \max_{j \in [m]} \Gamma_{i,j}^{t+1} \leq \sum_{i \in [n]} \sum_{j \in [m]} \Gamma_{i,j}^t L_j(\theta_j^{t+1}) - \tau \sum_{i \in [n]} \max_{j \in [m]} \Gamma_{i,j}^t.$$

We then have

$$\sum_{i \in [n]} \sum_{j \in [m]} \left( \Gamma_{i,j}^{t+1} L_i(\theta_j^{t+1}) - \Gamma_{i,j}^t L_i(\theta_j^t) \right) \tag{39}$$

$$\leq \sum_{i \in [n]} \sum_{j \in [m]} \Gamma_{i,j}^t \left( L_i(\theta_j^{t+1}) - L_i(\theta_j^t) \right) + \tau \sum_{i \in [n]} \max_{j \in [m]} \Gamma_{i,j}^{t+1} - \tau \sum_{i \in [n]} \max_{j \in [m]} \Gamma_{i,j}^t \tag{40}$$

$$\leq -\frac{\eta(1 - \nu\eta)}{2} \sum_{j \in [m]} \sum_{i \in [n]} \Gamma_{i,j}^t \|d_j^t\|^2 + \tau \sum_{i \in [n]} \max_{j \in [m]} \Gamma_{i,j}^{t+1} - \tau \sum_{i \in [n]} \max_{j \in [m]} \Gamma_{i,j}^t \quad \text{(apply Eq. 38)} \tag{41}$$

$$= -\frac{\eta(1 - \nu\eta)}{2} \sum_{j \in [m]} \beta_j \|d_j^t\|^2 + \tau \sum_{i \in [n]} \max_{j \in [m]} \Gamma_{i,j}^{t+1} - \tau \sum_{i \in [n]} \max_{j \in [m]} \Gamma_{i,j}^t. \tag{42}$$

Here $\beta_j$ is the $j$-th dimension of $\beta \in \Delta^m$ in Eq. 3. Hence,

$$\sum_{j \in [m]} \beta_j \|d_j^t\|^2 \leq \frac{2}{\eta(1 - \nu\eta)} \sum_{i \in [n]} \sum_{j \in [m]} \left( \Gamma_{i,j}^t L_i(\theta_j^t) - \Gamma_{i,j}^{t+1} L_i(\theta_j^{t+1}) \right) \tag{43}$$

$$+ \frac{2}{\eta(1 - \nu\eta)} \left( \tau \sum_{i \in [n]} \max_{j \in [m]} \Gamma_{i,j}^{t+1} - \tau \sum_{i \in [n]} \max_{j \in [m]} \Gamma_{i,j}^t \right). \tag{44}$$

Applying telescope sum for $t \in [T]$ on both sides gives

$$\sum_{t \in [T]} \sum_{j \in [m]} \beta_j \|d_j^t\|^2 \leq \frac{2}{\eta(1 - \nu\eta)} \sum_{i \in [n]} \sum_{j \in [m]} \left( \Gamma_{i,j}^1 L_i(\theta_j^1) - \Gamma_{i,j}^{T+1} L_i(\theta_j^{T+1}) \right) \tag{45}$$

$$+ \frac{2}{\eta(1 - \nu\eta)} \left( \tau \sum_{i \in [n]} \max_{j \in [m]} \Gamma_{i,j}^{T+1} - \tau \sum_{i \in [n]} \max_{j \in [m]} \Gamma_{i,j}^1 \right) := C. \tag{46}$$

Then we get the following bound on the average gradient norm:

$$\frac{1}{T} \sum_{t \in [T]} \sum_{j \in [m]} \beta_j \|d_j^t\|^2 \leq \frac{C}{T}. \tag{47}$$

If we take $\nu\eta = \frac{1}{2}$, then $C = O(\nu)$. This gives us a $O\left(\frac{\nu}{T}\right)$ rate in terms of gradient norms for non-convex cases under a fixed learning rate.

For the case where $K > 1$, we have

$$\sum_{i \in [n]} \sum_{j \in [m]} \left( \Gamma_{i,j}^{t+1} L_i(\theta_j^{t+1}) - \Gamma_{i,j}^t L_i(\theta_j^t) \right) \tag{48}$$

$$\leq -\frac{\eta(1 - \nu\eta)}{2} \sum_{j \in [m]} \beta_j \sum_{k \in [K]} \left\| d_j^{t,k} \right\|^2 \tag{49}$$

$$\leq -\frac{\eta(1 - \nu\eta)}{2} \sum_{j \in [m]} \beta_j \left\| d_j^t \right\|^2, \tag{50}$$

where $k$ denotes the index for inner updates on model parameters fixing $\Gamma^t$. Similarly, we have the result

$$\frac{1}{T} \sum_{t \in [T]} \sum_{j \in [m]} \beta_j \left\| d_j^t \right\|^2 \leq \frac{C}{T}. \tag{51}$$

## A.3 EXPERIMENTS WITH $n \ll m$

**Evaluation Metrics.** For applications with few objectives (small $n$), we use the hypervolume measure, a widely used metric for evaluating the quality of MOO solutions and is a proxy of diversity (Zitzler & Thiele, 1999). Hypervolume is feasible to compute when the number of objectives is small. Given a solution set $S \subset \mathbb{R}^n$ and a set of reference points $r = [r_1, \ldots, r_n] \subset \mathbb{R}^n$, the Hypervolume of $S$ measures the region weakly dominated by $S$ and bounded above by $r$: $H(S) = \Lambda \left( \{ q \in \mathbb{R}^n \mid \exists p \in S : p \leq q \text{ and } q \leq r \} \right),$ where $\Lambda(\cdot)$ denotes the Lebesgue measure. To ensure fair calculation of it, the reference points are kept consistent across all algorithms for each dataset. These reference points are determined either by following the settings of previous studies or by setting them as the upper bounds of the objective values from all algorithms to be compared.

**Hyperparameters.** For the extended version of MosT (denoted as MosT-E) described in Section 3.2, we introduce additional hyperparameters $\alpha_1, \ldots, \alpha_n$, and $n'$ to handle the extension of existing objectives. The parameter $\alpha_i$ represents the positive shape parameter of the Dirichlet distribution, used to generate diverse objective weights, and $n'$ represents the number of extended objectives.

### A.3.1 TOY PROBLEMS

We demonstrate the effectiveness of MosT on a toy ZDT problem set. It is a popular MOO benchmark containing two objectives ($n = 2$) with oracle Pareto fronts (Zitzler et al., 2000). Specifically, we use ZDT-1, ZDT-2, and ZDT3, which are problems with 30 variables and exhibit convex, concave, and disconnected Pareto-optimal fronts, respectively. We compare MosT with two baselines described in Section 6.1, and two additional methods—Exact Pareto Optimization (EPO) based on different preference vectors (Mahapatra & Rajan, 2020), and SVGD based on stein variational gradient descent (Liu et al., 2021).

Table 5: MosT achieves higher Hypervolumes than the baselines on the ZDT bi-objective problem.

|       | MGDA | Linearization | SVGD | EPO | MosT |
|-------|------|---------------|------|-----|------|
| ZDT-1 | $4.02_{\pm 0.92}$ | $5.72_{\pm 0.01}$ | $5.54_{\pm 0.12}$ | $4.40_{\pm 0.01}$ | $\mathbf{5.87}_{\pm 0.00}$ |
| ZDT-2 | $4.63_{\pm 0.94}$ | $6.65_{\pm 0.00}$ | $6.65_{\pm 0.00}$ | $6.65_{\pm 0.00}$ | $\mathbf{6.88}_{\pm 0.00}$ |
| ZDT-3 | $4.53_{\pm 0.83}$ | $6.27_{\pm 0.02}$ | $5.77_{\pm 0.15}$ | $4.53_{\pm 0.68}$ | $\mathbf{6.39}_{\pm 0.03}$ |

We report the Hypervolumes of each method in Table 5 and visualize the obtained solutions alongside the entire Pareto-optimal fronts in Figure 4 for a more intuitive comparison. The results in Table 5 demonstrate that MosT achieves higher Hypervolumes, indicating its superior ability to generate more diverse solution sets that cover larger areas. Further analysis of the Pareto fronts reveals the following observations: 1) EPO and SVGD prioritize reducing one loss, potentially resulting in biased trade-offs, with SVGD lacking guaranteed convergence to Pareto-optimal solutions; 2) MGDA produces diverse solutions but fails to cover the entire Pareto-optimal fronts; 3) Linearization-based MOO is a competitive baseline with high Hypervolumes, but its solutions do not provide satisfactory

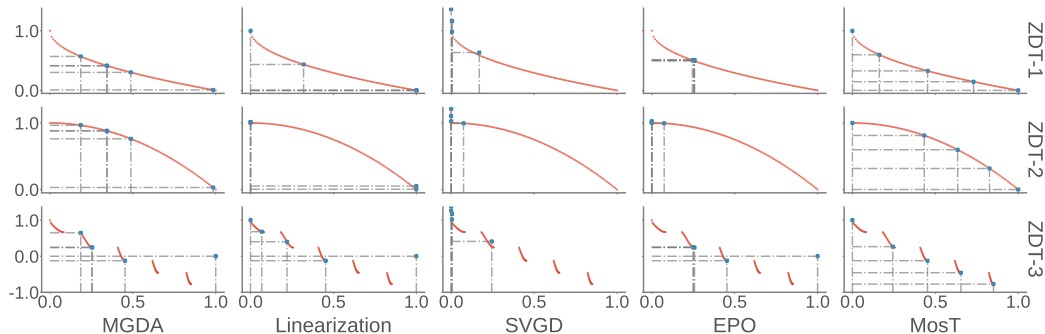

Figure 4: Solutions derived by different methods (blue scatters) on the ZDT bi-objective task, with the oracle Pareto-optimal fronts for the two objectives shown in red scatters.

diverse trade-offs, as evident from the Pareto fronts; 4) In contrast, MosT generates evenly-distributed solutions across the Pareto fronts.

**Investigation for EPO and SVGD.** Despite adhering to the official implementation of EPO, it fails to meet performance expectations due to its extensive requirement for preference vector sampling. Increasing the number of solutions ($m$) generated by EPO yields noticeable enhancements, particularly in ZDT-3 and, to a lesser extent, in ZDT-1, as shown in Table 6. However, even with a larger $m$, EPO consistently lags behind MosT, which achieves well-distributed solutions across Pareto fronts without relying on extensive sampling.

It is worth noting that we focus on scenarios where computational constraints limit us to training only $m$ models while needing to address numerous objectives ($n \gg m$). Therefore, we prioritize algorithms that afford better control over the number of generated solutions to achieve a promising trade-off.

Table 6: EPO's performance improves with larger $m$, yet still falls short compared to MosT, which achieves superior results with fewer $m$.

|  | EPO ($m = 5$) | EPO ($m = 100$) | MosT ($m = 5$) |
|---|---|---|---|
| ZDT-1 | $4.40_{\pm 0.01}$ | $4.46_{\pm 0.01}$ | $\mathbf{5.87}_{\pm 0.00}$ |
| ZDT-2 | $6.65_{\pm 0.00}$ | $6.65_{\pm 0.00}$ | $\mathbf{6.88}_{\pm 0.00}$ |
| ZDT-3 | $4.53_{\pm 0.68}$ | $6.09_{\pm 0.00}$ | $\mathbf{6.39}_{\pm 0.03}$ |

Additionally, we conduct more experiments on SVGD with a much larger number of solutions $m$ (i.e., intentionally being unfair to our method). We observe that (a) using a larger $m$ can improve its performance, but (b) even with $m = 100$, SVGD still cannot outperform MosT (ours) with $m = 5$.

Table 7: SVGD's performance improves with larger $m$, yet still falls short compared to MosT, which achieves superior results with fewer $m$.

|  | SVGD ($m = 5$) | SVGD ($m = 100$) | MosT ($m = 5$) |
|---|---|---|---|
| ZDT-1 | $5.54_{\pm 0.12}$ | $5.73_{\pm 0.03}$ | $5.87_{\pm 0.00}$ |
| ZDT-2 | $6.65_{\pm 0.00}$ | $6.66_{\pm 0.00}$ | $6.88_{\pm 0.00}$ |
| ZDT-3 | $5.77_{\pm 0.15}$ | $6.01_{\pm 0.02}$ | $6.39_{\pm 0.03}$ |

### A.3.2 FAIRNESS-ACCURACY TRADE-OFFS ($n \ll m$)

In this section, we apply MosT to explore various trade-offs between accuracy and algorithmic fairness (i.e., statistical independence between predictions and sensitive attributes). Thus, the number

of objectives $n$ is 2. However, in this scenario with limited number of objectives, using optimal transport to match solutions and objectives may produce feasible but trivial solutions as explained in Section 3.2. Hence, we adapt the extension of MosT named MosT-E (introduced in Section 3.2).

As discussed in Section 2, prior works that address fairness-accuracy trade-offs can be limited due to the difficulty of setting constraints before training (Zafar et al., 2017), or the mismatch between diverse exploration space and diverse solutions (Mahapatra & Rajan, 2020). MosT-E differs by sampling a wide range of preference vectors to encompass various trade-offs comprehensively and using optimal transport to automatically generate solutions that maximize coverage for all preference vectors. We quantify the fairness objective using *disparate impact* (Court, 1971), and optimize it using its convex approximation (Zafar et al., 2017). We experiment on a synthetic dataset (Zafar et al., 2017) and a real German credit dataset (Asuncion & Newman, 2007). We compare MosT and MosT-E with MGDA, linearization-based MOO (which can be viewed as a soft version of Zafar et al. (2017), and EPO (Mahapatra & Rajan, 2020), and select the best parameters for each method based on the highest Hypervolume on a validation set.

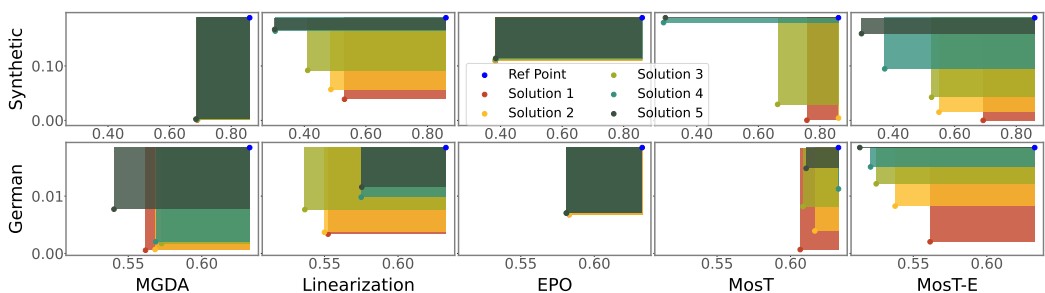

Figure 5: Hypervolumes (colored areas) formed by five solutions for classification loss (objective 1, x-axis) and fairness (objective 2, y-axis) on synthetic and German datasets.

Table 8: Hypervolumes ($\times 100$) on 5 solutions with different fairness-accuracy trade-offs. MosT-E achieves the highest Hypervolume coverage on two (fairness, accuracy) objectives.

|  | MGDA | Linearization | EPO | MosT | MosT-E |
|---|---|---|---|---|---|
| Synthetic | $3.28_{\pm 0.05}$ | $6.70_{\pm 0.04}$ | $2.44_{\pm 0.01}$ | $3.78_{\pm 0.04}$ | $\mathbf{7.65}_{\pm 0.06}$ |
| German | $5.14_{\pm 0.06}$ | $4.99_{\pm 0.05}$ | $4.56_{\pm 0.02}$ | $4.64_{\pm 0.05}$ | $\mathbf{5.27}_{\pm 0.06}$ |

**MosT-E generates more diverse trade-offs.** Table 8 shows that MosT-E achieves the highest Hypervolumes, suggesting a superior quality of the solution set it generates. Furthermore, Figure 5 demonstrates that MosT-E generates solutions that are not only more diverse but also more evenly distributed across the Pareto fronts.

**MosT-E effectively addresses the problem of MosT under $n \ll m$.** When $n \ll m$, MosT may assign models separately to dominate individual objectives, resulting in solutions without sufficient diversity. The solutions generated by MosT shown in Figure 5, align with our idea by predominantly prioritizing either low classification loss or low disparate impact. This limitation is effectively overcome by MosT-E, with diversely combining existing few objectives as new objectives.

## A.4 EXPERIMENTAL DETAILS

We will detail the models and hyperparameters used for each dataset. All algorithms follow the same setup, including the train-validation-test split, number of training epochs, and tunable learning rates.

**Hyperparameters for baselines.** In addition to the standard setup, we fine-tune hyperparameters specific to each baseline model, aligning with their original configurations. For instance, we adjust the hyperparameter responsible for scaling the proximal term in FedProx according to the recommendations provided in (Li et al., 2020).

**Description of KL divergence used.** We employ KL divergence to assess differences in $\Gamma$ across iterations and solution diversity. Specifically, we utilize symmetric KL divergence, defined as $\frac{(KL(P||Q)+KL(Q||P))}{2}$, where $Q$ and $P$ represent probabilities in the distribution of $P$ and $Q$, respectively.

### A.4.1 EXPERIMENTAL SETUP FOR TOY PROBLEMS

**ZDT bi-objective problems** (Zitzler et al., 2000). It contains a class of benchmark problems commonly used to evaluate optimization algorithms, particularly those designed for multi-objective optimization. These problems involve optimizing two conflicting objectives simultaneously. We specifically employ ZDT-1, ZDT-2, and ZDT-3 to evaluate the performance of algorithms. We use multinomial logistic regression, maintaining a consistent learning rate of 0.005 throughout training. This configuration aligns with the established setup presented in (Liu et al., 2021). We run 1,000 epochs for the datasets.

### A.4.2 EXPERIMENTAL SETUP FOR FEDERATED LEARNING

**Synthetic data** (Li et al., 2020). This synthetic dataset is specifically designed to provide controlled complexities and diverse scenarios for assessing the performance of algorithms. The synthetic data generation process relies on two hyperparameters, $\rho_1$ and $\rho_2$, which shape the dataset's characteristics. $\rho_1$ controls the heterogeneity among local models used to generate labels on each device. While $\rho_2$ governs the differences in data distribution among devices. Larger $\rho_1$ or $\rho_2$ introduces more heterogeneity. For the generated dataset, we conduct our experiments using a train-validation-test split ratio of 6:2:2. We use multinomial logistic regression as the model and run 400 epochs in total. The learning rates are swept from $\{0.005, 0.01, 0.05, 0.1\}$ without decaying throughout the training process.

**FEMNIST** (Cohen et al., 2017; Caldas et al., 2018). In addition to the synthetic datasets, we also conduct experiments on the Federated Extended MNIST (FEMNIST) dataset, a widely used real-world dataset in federated learning research (Li et al., 2020), using multinomial logistic regression. It comprises handwritten digit images from multiple users, encompassing 62 classes, including digits (0-9) and uppercase and lowercase letters (A-Z, a-z). The data is distributed across 206 clients, with each client holding a subset of the digit classes. This distribution simulates a real-world federated learning scenario, prioritizing data privacy and distribution concerns. We employ a convolutional neural network featuring two convolutional layers with ReLU activation, followed by max-pooling. Additionally, a fully connected layer maps the flattened features to 62 output classes. We run 400 epochs for training. Learning rates are swept from $\{0.08, 0.1\}$.

### A.4.3 EXPERIMENTAL SETUP FOR FAIRNESS-ACCURACY TRADE-OFF

In the context of fairness-accuracy trade-off, we experiment on two datasets, the synthetic dataset and the German dataset, introduced below. We employ multinomial logistic regression as our model, conducting 20 epochs of training and sweeping learning rates from $\{0.08, 0.1\}$. We use the enhanced MosT-E for the German dataset. MosT-E extends the existing $n$ objectives to $n'$ by interpolating them with weights drawn from a Dirichlet distribution. We set all shape parameters, $\alpha_1, \ldots, \alpha_n$, to the same value within the range $[0.1, 0.5, 1.0]$. The number of extended objectives is chosen from $[10, 15, 20]$. In practice, we observe that extending the original 2 objectives to 10 yields results similar to those obtained with 20 objectives.

**Synthetic dataset** (Zafar et al., 2017). The synthetic dataset contains 2,000 binary classification instances generated randomly as specified in (Zafar et al., 2017). Binary labels for classification are generated using a uniform distribution. It features 2-dimensional nonsensitive features generated from two distinct Gaussian distributions, and a 1-dimensional sensitive feature generated using a Bernoulli distribution.

**UCI German credit risk dataset** (Asuncion & Newman, 2007). This dataset comprises 1,000 entries, each characterized by 20 categorical and symbolic attributes. These attributes serve to classify individuals as either good or bad credit risks. Gender is considered as the sensitive attribute in this context.

### A.4.4 Experimental Setup for Multi-Task Learning

In the realm of multi-task learning, we assess the efficacy of MosT using the Office-Caltech10 and DomainNet datasets. The number of objectives $n$ varies across the datasets: $n = 4$ for Office-Caltech10 and $n = 6$ for DomainNet. We initialize the models with pre-trained weights for both datasets, leveraging ImageNet-pretrained ResNet-18 (He et al., 2016) for Office-Caltech10 and ConvNeXt-tiny (Liu et al., 2022b) for DomainNet.

It is worth noting that Pareto Multi-task Learning (PMTL) (Lin et al., 2019) is a notable method, but its exclusion from our comparison is due to concerns regarding computational efficiency, particularly when applied to large-scale real-world datasets.

**Office-Caltech10 dataset** (Saenko et al., 2010; Griffin et al., 2007). The Office-Caltech10 dataset comprises images from four distinct data sources: Office-31(Saenko et al., 2010) (three data sources) and Caltech-256 (Griffin et al., 2007) (one data source). These sources capture images using different camera devices or in various real environments with diverse backgrounds, representing different objectives.

**DomainNet dataset** (Peng et al., 2019). The DomainNet dataset includes natural images sourced from six distinct data sources: Clipart, Infograph, Painting, Quickdraw, Real, and Sketch. This dataset is characterized by its diversity, covering a wide range of object categories. For our experiments, we focus on a sub-dataset composed of the top ten most common object categories from the extensive pool of 345 categories within DomainNet, following (Li et al., 2021).

### A.4.5 Experimental Setup for Prompt Learning

We explore prompt learning across three datasets from the SuperGLUE benchmark: **BoolQ** (Clark et al., 2019), **MultiRC** (Khashabi et al., 2018), and **WiC** (Pilehvar & Camacho-Collados, 2018). In our approach, each instance represents a distinct objective. This framework allows us to delve into prompt learning using a limited set of training instances while aiming for generalization to unseen test instances. We randomly sample 128 instances from the training dataset and evenly partition the original validation dataset to form both the validation and test datasets. Our training involves a soft prompt approach based on the T5-base model, with the base model parameters kept frozen. Parameter setup follows (Qin & Eisner, 2021).

For prompt learning, where each instance is considered an objective, the absence of client groups or task types, as seen in federated learning or multi-task learning, prevents us from evaluating solution performance over the validation set and then selecting the best solution for inference. To address this, we train a simple dispatcher to learn the correlation between instances and solutions (prompts), predicting the optimal solution for a given instance. Specifically, we train cross-attention on the hidden embedding of soft prompts and instances, with architecture following (Lee et al., 2018) (Section 3.1). These hidden embeddings are generated from a fixed encoder of the T5-base.

### A.5 Ablation Study on MosT Design

| | MGDA | Linearization | FedAvg | FedProx | FedMGDA+ | MosT w/o $R(\Gamma)$ |
|---|---|---|---|---|---|---|
| Syn (0.0, 0.0) | $77.22_{\pm0.41}$ | $75.91_{\pm0.37}$ | $75.71_{\pm0.51}$ | $75.60_{\pm0.42}$ | $75.26_{\pm1.21}$ | $83.09_{\pm0.87}$ |
| Syn (0.5, 0.5) | $87.09_{\pm0.29}$ | $87.18_{\pm0.27}$ | $86.26_{\pm0.61}$ | $86.13_{\pm0.39}$ | $85.21_{\pm1.42}$ | $89.07_{\pm0.63}$ |
| Syn (1.0, 1.0) | $90.52_{\pm0.13}$ | $89.87_{\pm0.51}$ | $88.12_{\pm0.75}$ | $87.58_{\pm1.36}$ | $87.16_{\pm1.09}$ | $91.70_{\pm0.02}$ |

| MosT (O) | MosT (M) | MosT (M, soft) | MosT w/o CL | w-MGDA | MosT (L) | MosT |
|---|---|---|---|---|---|---|
| $76.65_{\pm0.81}$ | $67.62_{\pm3.46}$ | $74.66_{\pm0.70}$ | $81.97_{\pm0.58}$ | $76.80_{\pm0.79}$ | $82.62_{\pm0.33}$ | $\mathbf{84.25}_{\pm0.51}$ |
| $86.94_{\pm0.61}$ | $78.98_{\pm2.04}$ | $80.15_{\pm1.54}$ | $88.19_{\pm0.40}$ | $86.18_{\pm1.19}$ | $88.85_{\pm0.34}$ | $\mathbf{89.99}_{\pm0.52}$ |
| $90.42_{\pm0.22}$ | $75.20_{\pm3.75}$ | $73.20_{\pm4.71}$ | $91.25_{\pm0.51}$ | $89.32_{\pm0.76}$ | $91.26_{\pm0.44}$ | $\mathbf{92.21}_{\pm0.08}$ |

To generate diverse and complementary solutions for multiple objectives, MosT first finds a balanced matching between objectives and solutions by OT along with elaborated learning strategies and then locates the descent direction common to re-weighted objectives using MGDA. In this section, we conduct comprehensive ablation studies to verify the MosT design. Specifically, for OT, we verify the necessity of it (Appendix A.5.1) and its specific designs, including solution-specialization curriculum (Appendix A.5.2) and sparsity encouragement imposed by L1 regularization (Appendix A.5.3). Additionally, we evaluate the necessity of MGDA in Appendix A.5.4. These ablation studies contribute to a thorough understanding of the effectiveness of each component within the overall MosT framework. Experiments are carried out on three synthetic federated learning datasets, with results shown in Table A.5 and Figure 6.

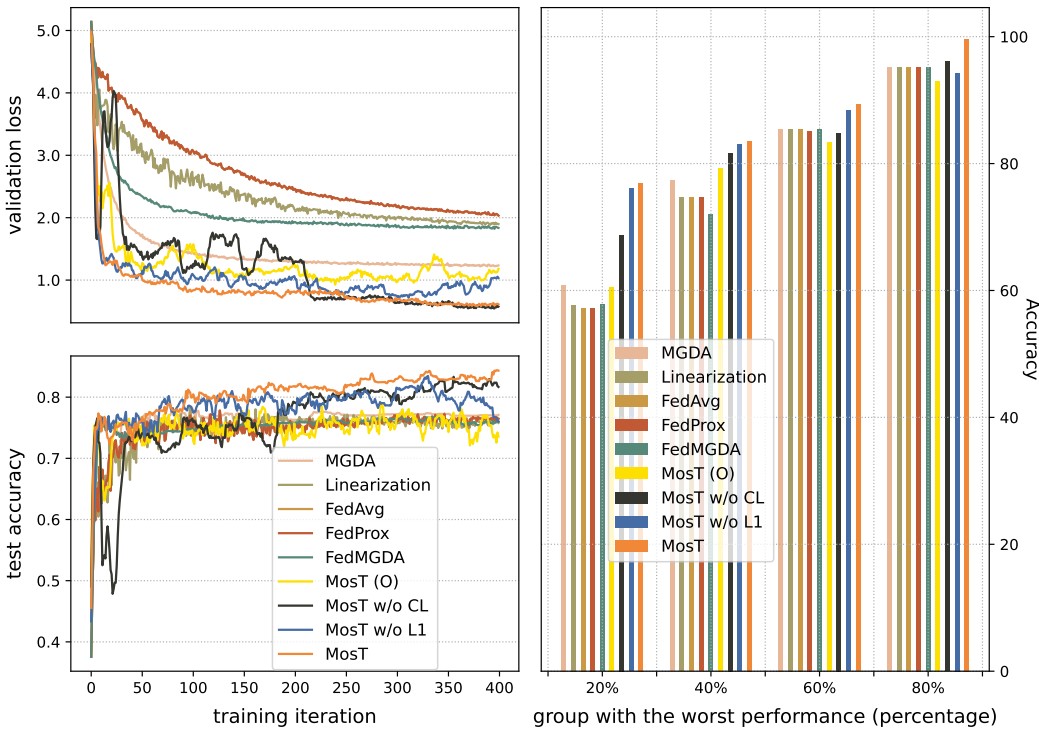

Figure 6: Including MosT with a series of ablation study results, (a) displays training loss and test accuracy curves; (b) shows the accuracy of the worst 20%, 40%, 60% and 80% client groups. We omitted MosT (M) from panel for clarity, as it exhibits inferior performance compared to other methods.

### A.5.1 ABLATION STUDY FOR OT

**OT-Generated v.s. Randomly Generated Weight Assignments.** We compare the weight assignments generated by OT with the randomly generated weight assignments. This can verify the impact of the choice of objective weighting method in MGDA on the overall performance of MosT. In other words, we compare MosT with executing MGDA $m$ times by using randomly generated weights to reweight objectives, which is denoted as *w-MGDA*. Experimental results reveal that OT-generated weights work significantly better than random weights. *This illustrates the necessity of using OT to find a balanced match between solutions and objectives.*

**Different Matching Strategies.** We also conduct ablation studies on the effectiveness of the optimal transport matching (Eq. 3) by comparing three strategies introduced in Section 3.3: 1) the original MosT objective, which utilizes optimal transport; 2) "objective selecting model", which selects the best expert/model for each objective (i.e., removing the $\Gamma^\top \mathbf{1}_n = \beta$ constraint); and 3) "model selecting objective", which selects the best objective for each model (i.e., removing the $\Gamma \mathbf{1}_m = \alpha$ constraint). These two variants are denoted as *MosT (O)* and *MosT (M)*, resp., with results shown

in Table A.5 and Figure 6. We also try a 'soft' version of *MosT (M)*, utilizing normalized loss over objectives when training each model, denoted as *MosT (M, soft)*.

To ensure a fair comparison, we initialize comparisons with the same model weights. Throughout the training process, we track the assignment of objectives to each model, i.e., for every model, identifying the objectives with the smallest validation loss. We visualize the percentages of the selected objectives for each model over time in Figure 7 on the Syn (0.0, 0.0) dataset. In the case of "objective selecting model" (middle), we observe that two of the models progressively dominate all the objectives. Similarly, "model selecting objective" shows the early dominance of one model. While its soft version, MosT (M, soft), exhibits some improvement over MosT (M), it still lags behind other methods because of the imbalanced training over the objectives. These findings emphasize the necessity of achieving a balanced model-objective trade-off. These observations confirm the presence of the collapse phenomenon in MOO, where limited solutions dominate all objectives. On the contrary, MosT using optimal transport involving a two-way matching shows a more balanced distribution of objectives among the models throughout training.

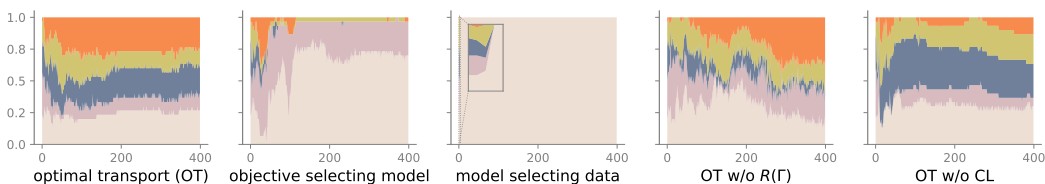

Figure 7: The percentage of assigned objectives for each model under three matching strategies and two variants of MosT. Each color band represents a model, with the y-axis indicating the corresponding percentage. We see that MosT (leftmost) learns 5 diverse models that serve the 30 objectives in a balanced manner.

### A.5.2  ABLATION STUDY FOR OT DESIGN - CURRICULUM LEARNING

We evaluate the impact of curriculum learning (from Section 3.3) on optimizing multiple models using MosT.

**Curriculum setup for MosT.** As introduced in Section 3.3, our proposed curriculum strategy involves adjusting marginal distributions $\alpha$ and $\beta$ over different training stages to balance the freedom of 'model selecting objective' and 'objective selecting model'. In the initial stages, we prioritize a uniform distribution for $\beta$ to ensure exposure to multiple objectives. As training progresses, we transition $\alpha$ to a uniform distribution, covering all objectives, while relaxing $\beta$. This transition is achieved by a hyperparameter that gradually decreases from 1 to 0. Though this hyperparameter gradually approaches zero, the transition direction differs: it shifts $\beta$ from uniform to performance-oriented and, conversely, shifts $\alpha$ in the opposite direction.

We compare standard MosT with a variant using uniform marginal distributions for $\alpha$ and $\beta$ throughout training, denoted as *MosT w/o CL*. We hypothesize that curriculum learning enhances overall performance and training stability. We conduct experiments on three synthetic federated learning datasets. The results in Table A.5 show that using curriculum learning significantly improves the performance of MosT, proving its effectiveness. Notably, even without curriculum learning, MosT outperforms other algorithms. Furthermore, Figure 6 illustrates the training loss and test accuracy curves, highlighting the stability difference between the two approaches during training. Curriculum learning leads to increased stability and better convergence towards better solutions.

### A.5.3  ABLATION STUDY FOR OT DESIGN - DIVERSITY ENCOURAGEMENT

As detailed in Section 4 and supported by empirical evidence in Section 5, encouraging a sparse and balanced alignment between objectives and solutions leads to solution specialization on objectives. MosT goes a step further by employing diversity regularized optimal transport to promote diversity.

**Enhancing Diversity through Sparse Transport and Regularization.** Before imposing diversity regularization to further enforce diversity, we aim to verify whether MosT without diversity regular-

ization can generate diverse solutions. We track solution diversity throughout the training process using KL divergence, as explained in Section 6.2, with results depicted in Figure 9. Our empirical findings indicate that even without $R(\Gamma)$, sparse transport yields diverse solutions that balance all objectives. However, by setting $R(\Gamma)$ to be the negative of our diversity measure (Definition 1), we can further encourage diversity. In Figure 8, we depict the distribution of model specialization across objectives, assessed through normalized model accuracies. Notably, solutions trained with MosT exhibit a tendency to specialize in specific objectives, underscoring their heightened diversity compared to baseline approaches. And these diverse solutions then jointly perform better.

**Performance Impact of Diversity Encouragement.** Building on the motivation outlined in Section 5, this section focuses on showcasing the performance benefits resulting from diversity encouragement. To assess the impact of diversity regularization, we conduct a comparative analysis between scenarios with and without it. Table A.5 and Figure 6 highlight that MosT with diversity regularization not only enhances performance but also contributes to the fairness of federated learning.

### A.5.4 ABLATION STUDY FOR MGDA

**MGDA v.s. Linearization in Weighted Multi-Objective Optimization.** We compare MosT that uses MGDA against the variant that updates model parameters based on the optimal transport solution weights. We aim to understand how effectively these two methods determine gradient updates for weighted multi-objective optimization. In this variant of MosT, denoted as *MosT (L)*, instead of seeking the Pareto solution of $m$ weighted objectives (as indicated in Eq. 1), we compute $\theta_j$ as $\theta_j = \sum_{i=1}^n \Gamma_j^i \theta_j^i$, where $\theta_j^i$ represents the parameter of $\theta_j$ trained on data from the $i$-th objective. The experimental results showcase the consistent superiority of MosT over both MGDA and *MosT (L)* across three synthetic federated learning scenarios. *It proves the effectiveness of employing MGDA for parameter updates.*

### A.6 COMPARATIVE RUNTIME ANALYSIS

We assess the runtime of algorithms on the same platform, providing analyses for various applications: federated learning (Table 9), multi-task learning (Table 10), mixture-of-prompt learning (Table 11), ZDT datasets (Table 12), and fairness-accuracy trade-off (Table 13). Additionally, we include the computation time required for OT and MGDA within MosT.

Our results indicate that MosT exhibits comparable running times to baselines, despite MosT involving the computation of OT and MGDA, both of which only account for negligible time. However, for MosT-E, which explicitly extends the number of objectives, it will require more time than baselines.

Additionally, we provide a **detailed decomposition on the complexity of all the components** below.

At each round, we alternate between the optimization of $\Gamma$ and solving a reweighted MGDA problem. Utilizing IPOT, the complexity for optimizing $\Gamma$ (Equation 5) is roughly $O(mn)$ [1]. Optimizing $\theta_{1:m}$ using reweighted MGDA (Equations 7 and 8) requires $n$ gradient access for each solution $\theta_j$ ($j \in [m]$), which takes up $O(mn)$ time in total. Hence, The overall complexity of our algorithm per iteration is $O(mn)$. When $m \ll n$ (our mainly focused setting), e.g., $m = O(\log n)$, the above complexity can reduce to $O(n \log n)$. Note that though the OT step and MGDA step both scale similarly, their actual running time is drastically different as MGDA requires gradient computation.

Table 9: Runtime (sec) comparisons for all methods on federated learning datasets, performed on a single Nvidia RTX A5000 platform.

|  | MGDA | Linearization | FedAvg | FedProx | FedMGDA+ | MosT | MosT-OT | MosT-MGDA |
|---|---|---|---|---|---|---|---|---|
| Syn (0.0, 0.0) | 219.86 | 225.90 | 222.22 | 281.69 | 516.25 | 217.59 | 1.00 | 0.44 |
| Syn (0.5, 0.5) | 208.82 | 208.27 | 205.90 | 258.19 | 495.62 | 201.70 | 0.92 | 0.23 |
| Syn (1.0, 1.0) | 269.44 | 268.33 | 270.65 | 333.74 | 557.58 | 260.50 | 0.98 | 0.35 |
| FEMNIST | 3522.83 | 3135.76 | 3147.62 | 3539.85 | > 5000.00 | 3368.63 | 0.94 | 45.35 |

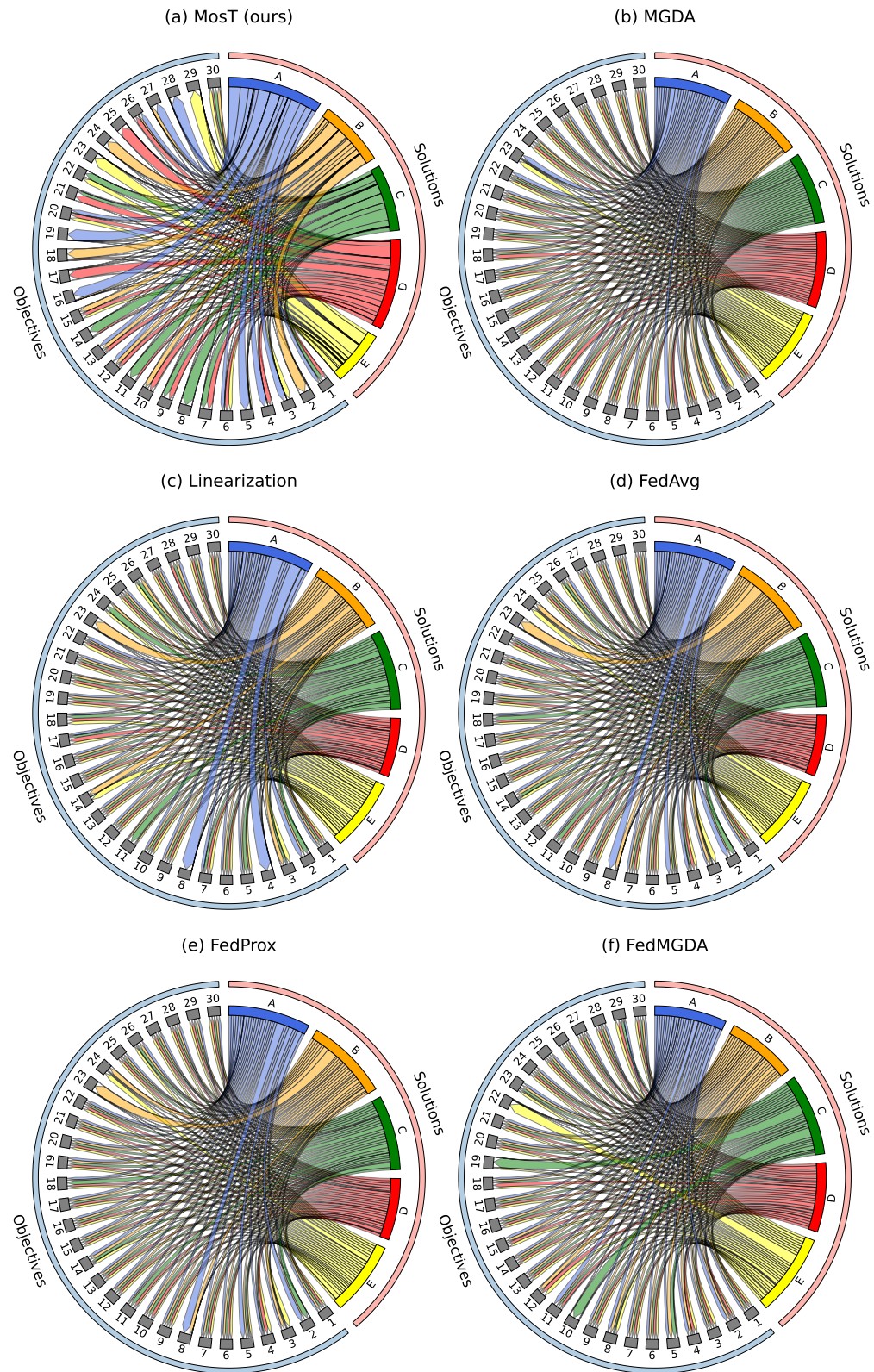

Figure 8: Proportion of 5 solutions (A/B/C/D/E), trained by MosT (a) and baselines (b)-(f), special-ized on 30 objectives (labeled as numbers). Notably, in (a), wider ribbons indicate that MosT-trained solutions address each objective with more specialization. In contrast, baseline solutions exhibit similar specializations across objectives. This highlights the enhanced solution diversity of MosT, a key factor contributing to its overall performance as shown in Figure 1.

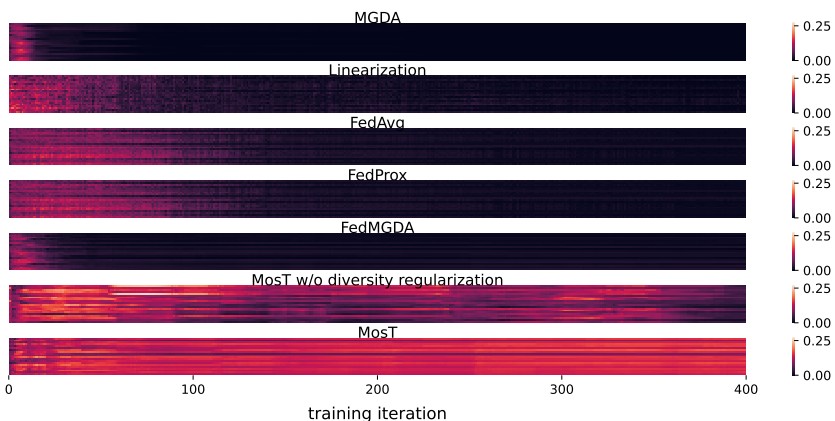

Figure 9: KL divergence between pairwise solution predictions. Baselines show decreasing diversity over training iterations, whereas both MosT variants maintain diversity. MosT with diversity regularization fosters diversity.

Table 10: Runtime (sec) comparisons for all methods on multi-task learning datasets, performed on a single Nvidia RTX A5000 platform.

|  | MGDA | Linearization | EPO | MosT | MosT-OT | MosT-MGDA |
|---|---|---|---|---|---|---|
| Office-Caltech10 | 465.73 | 669.88 | 775.24 | 371.06 | 0.54 | 8.08 |
| DomainNet | 294.23 | 341.27 | 347.23 | 226.9 | 0.08 | 15.43 |

Table 11: Runtime (sec) comparisons for all methods on prompt learning datasets, performed on a single Nvidia RTX A5000 platform.

|  | MGDA | Linearization | MosT | MosT-OT | MosT-MGDA |
|---|---|---|---|---|---|
| BoolQ | 2287.03 | 2170.97 | 1568.47 | 0.29 | 0.02 |
| MultiRC | 2108.14 | 1820.73 | 1892.47 | 0.37 | 0.09 |
| WiC | 1286.30 | 1187.56 | 1254.51 | 0.37 | 0.05 |

Table 12: Runtime (sec) comparisons for all methods on ZDT datasets, performed on a single Nvidia RTX A4000 platform.

|  | MGDA | Linearization | SVGD | EPO | MosT | MosT-OT | MosT-MGDA |
|---|---|---|---|---|---|---|---|
| ZDT-1 | 123.77 | 13.03 | 9.26 | 34.36 | 38.05 | 1.07 | 2.19 |
| ZDT-2 | 124.02 | 13.32 | 9.29 | 34.54 | 37.32 | 0.77 | 1.43 |
| ZDT-3 | 140.85 | 14.98 | 10.08 | 37.75 | 40.82 | 0.87 | 1.99 |

Table 13: Runtime (sec) comparisons for all methods on fairness-accuracy trade-off datasets, performed on a single Nvidia RTX A4000 platform.

|  | MGDA | Linearization | EPO | MosT-E | MosT-OT | MosT-MGDA |
|---|---|---|---|---|---|---|
| Synthetic | 888.59 | 43.32 | 79.09 | 1159.17 | 0.15 | 0.08 |
| German | 454.64 | 23.98 | 43.10 | 595.34 | 0.14 | 0.03 |

