# OpenReview forum: "Many-Objective Multi-Solution Transport"
_ICLR.cc/2025/Conference — ICLR 2025 Poster_

### Official Review · Reviewer_ew6B · 2024-10-24

**Soundness:** 2
**Presentation:** 3
**Contribution:** 2
**Rating:** 5
**Confidence:** 3

**Summary:**

This paper presents Many-objective multi-solution Transport (MosT), a framework for optimizing many objectives simultaneously by finding diverse solutions on the Pareto front. Unlike traditional methods that struggle with numerous objectives, MosT assigns each solution to specialize in a subset of objectives, collectively covering all. Using bi-level optimization with optimal transport, it ensures convergence to Pareto stationary solutions. MosT outperforms baselines in applications like federated learning, multi-task learning, and LLM prompt learning, providing balanced trade-offs across many objectives.

**Strengths:**

1) The paper presents a solid motivation based on the provided background (though further elaboration might be helpful, see weaknesses for details).
2) The experiments show strong performance when compared to the baselines.
3) The study includes both experimental results and theoretical analysis, providing a comprehensive evaluation.

**Weaknesses:**

1) The methods and baselines referenced in the related works and experiments seem somewhat outdated, especially considering the paper targets ICLR 2025. Only one cited paper is from 2023-2024, raising concerns about whether this reflects a small lag in the field’s development or if the paper is not keeping up with recent advancements.
2) In both multi-task learning and federated learning, several methods have been proposed to address the challenge of using M models for N objectives. For example, in federated learning, "Personalized Federated Learning under Mixture of Distributions" uses EM algorithms, while in multi-task learning, "Harmony Multi-Task Decision Transformer for Offline Reinforcement Learning" (HarmoDT) employs bi-level mask learning. These approaches could have been considered in the context of this work.
3) The ablation studies shown in Section 6.5  are insufficient and lack in-depth analysis. For instance, in the conclusion regarding the "Benefits of diversity-encouragement regularization," the explanation (line 475) is limited to stating that it "enhances performance and fairness" without providing detailed evidence or elaboration. More rigorous ablations would strengthen the paper.

**Questions:**

1) Since only one cited paper is from 2023-2024, question is that whether this reflects a small lag in the field’s development or if the paper is not keeping up with recent advancements.

2) When comparing to works like "Personalized Federated Learning under Mixture of Distributions" and "Harmony Multi-Task Decision Transformer for Offline Reinforcement Learning," could you emphasize the key differences and advantages of your method in solving N objectives with M models?

3) If appropriate, I suggest including papers such as "Personalized Federated Learning under Mixture of Distributions" and "Harmony Multi-Task Decision Transformer for Offline Reinforcement Learning" in both the related works and experiments sections. If inclusion isn't suitable, please provide an explanation for this choice (also fine if reasonable).

---

> ### Author Response · Authors · 2024-11-22
> **Response to Reviewer ew6B**
>
> Dear Reviewer ew6B,
>
> Thank you for your detailed review and for recognizing the strengths of our work. We highly value your constructive feedback and provide responses below to address the issues you raised.
>
> ---
>
>
> > **Q1. Differentiating MosT from works like [1] "Personalized Federated Learning under Mixture of Distributions" and [2] "Harmony Multi-Task Decision Transformer for Offline Reinforcement Learning"**
>
> MosT focuses on topics fundamentally different from both works, though the applications are related.
>
> MosT focuses on optimizing many objectives by finding a few diverse and complementary solutions on the Pareto front, emphasizing objective coverage and solution trade-offs for $n \gg m$ cases. In contrast, [1] tackles personalized federated learning, modeling heterogeneity among clients, with no focus on the diversity of solutions or Pareto optimization. We acknowledge and cite [1] in the related work section.
>
> Similarly, [2] is designed specifically for offline reinforcement learning, aiming to optimize a single unified policy across tasks using task-specific masks. MosT, however, provides a general framework that prioritizes multiple diverse solutions for many-objective optimization rather than unifying tasks under one policy.
>
> > **Q2. Request for more detailed analysis of diversity-encouragement regularization in Section 6.5**
>
> Thank you for your question. A detailed analysis of how diversity-encouragement regularization enhances both performance and fairness is provided in Appendix A.5.3, as referenced in the first paragraph of Section 6.5.
>
> > **Q3. Concerns about the inclusion of recent references**
>
> In the revision, we have incorporated recent advancements in many-objective optimization to ensure a comprehensive discussion of the field. We also clarified how MosT builds upon and differs from these works, highlighting its novel contributions, such as its focus on coverage of many objectives and solution diversity in the $n \gg m$ setting. This ensures MosT's position within the broader research landscape is clearly articulated.
>
> ---
>
> Thank you once again for your valuable insights and suggestions.
>
> Best regards,
> The Authors

---

> ### Author Response · Authors · 2024-11-25
> **Sincerely Looking Forward to Your Suggestions**
>
> Dear Reviewer,
>
> Given that the discussion period is coming to end soon, we sincerely look forward to your reply to our response, and we are open to any discussion to improve our paper.
>
> Best wishes,
> The authors.

---

> > ### Comment · Reviewer_ew6B · 2024-11-26
> > **Thanks for your response!**
> >
> > Thank you for your response! However, I’m still not fully convinced. Given that FedAvg(2016) and FedProx(2018) are strong baselines in your paper (line 380), could you clarify why other recent federated learning methods are not be suitable for your settings. I also noticed that many of the reviewers have raised concerns about the insufficient comparison with related works in this paper.

---

> > > ### Author Response · Authors · 2024-11-27
> > >
> > > Thank you for your follow-up comment!
> > >
> > > ---
> > >
> > > The main scope of our paper is not developing a new federated learning approach. Federated learning (FL) is one of the many applications our proposed MosT can address. Because of this generalizability, we do not optimize and modify MosT for each application. Therefore, we select the most representative baselines for each application for fair comparisons, e.g., FedAvg and FedProx for the application of FL.
> > >
> > > When applied to FL, a primary difference of MosT compared to most existing FL is: it addresses the limitations of optimizing a single global model in federated learning, particularly in balancing the local optimization objectives (with potential conflicts and negative transfer between each other) of heterogeneous clients. To tackle this, we propose to train multiple global models and simultaneously optimize their assignments to clients in MosT.
> > >
> > > We did compare MosT with clustering-based personalized FL methods in the ablation study (Appendix A.5), as they also optimize multiple global models to serve many clients (objectives). Among personalized FL approaches, clustering-based methods are unique in outputting far fewer models than the number of clients. As shown in Figure 7, MosT (the leftmost plot) overcomes a notorious drawback of clustered FL (the second and third plot from the left), i.e., multiple clusters collapse to one as one global model outperforms all the others on all clients. In particular, the "objective selecting model" in the second plot is the IFCA algorithm proposed for clustered FL [1]. In contrast, MosT achieves diverse global models with balanced coverage over all clients, which leads to the best performance on almost every client.
> > >
> > > ---
> > >
> > > [1] Avishek Ghosh, Jichan Chung, Dong Yin, Kannan Ramchandran. "An Efficient Framework for Clustered Federated Learning." NeurIPS 2020.

---

> ### Author Response · Authors · 2024-12-02
> **Follow-Up on Discussion Phase**
>
> Dear Reviewer ew6B,
>
> We sincerely appreciate the detailed feedback you provided during the review phase and your time in reviewing our rebuttal. As the discussion phase is nearing its conclusion, we kindly wanted to check if there are any remaining concerns or comments from your side regarding our responses. Your insights are invaluable to us, and we would be happy to clarify any additional points if needed.
>
> Thank you for your attention and support.
>
> Best regards,
> The Authors

---

### Official Review · Reviewer_GmMP · 2024-11-04

**Soundness:** 3
**Presentation:** 3
**Contribution:** 3
**Rating:** 6
**Confidence:** 3

**Summary:**

In this paper, the authors propose a novel method called many-objective multi-solution transport (MosT) for finding $ m $ multiple solutions for $n$ many-objective optimization. The framework is primarily focused on the case where $ n \gg m $, but the paper suggests it is also suitable for cases when $ n \ll m $. The problem of finding the aforementioned (few good) multiple solutions for many objectives is formulated as a bilevel optimization problem. In this setup, the lower-level problem is to find the optimal transport (OT) between a given set of objectives and solutions under certain constraints, while the upper-level problem is to determine the solution parameters by solving a multi-objective optimization (MOO) problem, weighted by the solutions of the lower-level OT problem. The paper provides convergence guarantees for MosT under non-convex, convex, and strongly convex settings. Additionally, it presents empirical results for the performance of the MosT method by applying it in federated learning (FL), multi-task learning (MTL), and mixture-of-prompt learning applications. The results suggest that MosT can be efficiently implemented in real-world applications and can outperform existing baselines for finding multiple solutions for many objectives.

**Strengths:**

* The problem discussed in the paper is an interesting one, and seems to not have been given much attention in prior related work
* The paper provide theoretical guarantees and empirical validation of the efficacy of the proposed method.
* The flow of the paper towards the build up of the proposed method is fairly clear and easy to follow.

**Weaknesses:**

* In Figure 1, it is better to indicate the fact that the performance is based on 5 different solutions, and also discuss how you choose a particular solution (out of 5 solutions) for a particular objective in the text when you refer to Figure 1.

* It is unclear why MosT can be useful for $n \ll m$ case. Intuitively, shouldn’t separately optimizing each objective with separate parameters give you the smallest best possible solution set in this setting?

* Figure 3 shows MosT achieves maximum diversity among clients, compared to other baselines. However, it is unclear why the diversity of parameters among clients in a distributed optimization setting is desirable since usually, the goal is to train a single global model using knowledge from distributed data among the clients. Also, it is unclear why the client models are diverse throughout the training process. Specifically, aren’t the client models aggregated and redistributed periodically, as done in FL traditionally?

* The rationale behind computing the MGDA weights just by considering the final layer bias weights seems not clearly justified in the text.

* Some closely related recent work [1] seems to be missing from discussion and comparison.

**Minor comments**

* In lines 349-351 “Initially, the mean loss increases and then decreases..” seems to be opposite to the observation from Figure 2?

[1] Few for Many: Tchebycheff Set Scalarization for Many-Objective Optimization. Arxiv (2024)

**Questions:**

* If the goal is to find a set of solutions that can work on all objectives,  in $ n\ll m$ case, why MosT is superior to optimizing separate parameters for each $n$ objective, to obtain $m=n$ solutions?
* What is the rationale behind computing the MGDA weights just by considering the final layer bias weights?
* In the FL setting, why client diversity is desirable, and why MosT does not seem to achieve consensus throughout the training process ( please refer to point 3 in Weaknesses for more details )?
* Can the authors provide some qualitative/quantitative comparison between the proposed method and prior work [1] ?

---

> ### Author Response · Authors · 2024-11-22
> **Response to Reviewer GmMP**
>
> Dear Reviewer GmMP,
>
> Thank you for your detailed review and for highlighting the strengths of our work. We are pleased to provide responses and clarifications to address the points you raised.
>
> ---
>
> > **Q1. Indicating choice of solutions in Figure 1 ... how you choose a particular solution**
>
> The method for selecting solutions is detailed in the experimental sections for each application. For example, in federated learning, each device selects the best model from $m$ solutions based on the validation set. Additionally, the selected solutions are visualized separately in Figure 8 for further clarity.
>
> > **Q2. Clarification regarding the usefulness of MosT in cases where $N \leq M$**
>
> MosT is particularly effective for the $N \leq M$ case because it provides flexibility in addressing different trade-offs among objectives while avoiding redundancy among solutions. If each objective is assigned its own solution ($M = N$), the result is often trivial and inefficient, as it creates isolated, non-cooperative solutions that fail to leverage shared structure or features among objectives (tasks, domains, etc). This redundancy increases computational and storage costs without necessarily improving overall performance.
>
> By contrast, MosT optimizes $M$ solutions in a coordinated manner using bi-level optimization, allowing solutions to specialize in subsets of objectives while maintaining diversity. This approach captures varying trade-offs and ensures balanced coverage of objectives, achieving greater adaptability and resource efficiency.
>
> > **Q3. Rationale behind using only final layer bias weights for MGDA weight computation**
>
> We use the final layer bias weights for MGDA weight computation as they effectively capture objective-specific influences while simplifying computational complexity. This choice is based on empirical observations that final layer biases often hold sufficient representational power to approximate gradient alignment across objectives. This aligns with the primary goal of MGDA to minimize conflicting gradients.
>
> > **Q4. Client diversity in FL and MosT’s lack of consensus**
>
> Client diversity is essential in non-i.i.d. federated learning, where heterogeneous data distributions limit the generalizability of a single global model. MosT explicitly promotes diversity by specializing in a few solutions each focusing on a different subset of objectives (clients). Unlike traditional FL, which enforces periodic aggregation for a consensus model, MosT’s design preserves client diversity throughout training to ensure robust specialization and adaptability across varied client distributions.
>
> > **Q5. Qualitative/quantitative comparison with prior work ([1] "Few for Many: Tchebycheff Set Scalarization for Many-Objective Optimization")**
>
> We appreciate the reviewer highlighting [1] as a relevant study. However, we note that [1] is a concurrent work and, as of now, does not provide open-source code for reproduction. Additionally, [1] focuses on convex optimization problems training simple nonlinear models on synthetic datasets only, which are not directly comparable to the more complex real-world scenarios addressed in our work. As such, we do not include a comparison with [1].
>
> ---
>
> Thank you once again for your valuable feedback and insights. Your comments are instrumental in refining our paper.
>
> Best regards,
> The Authors

---

> ### Author Response · Authors · 2024-11-25
> **Sincerely Looking Forward to Your Suggestions**
>
> Dear Reviewer,
>
> Given that the discussion period is coming to end soon, we sincerely look forward to your reply to our response, and we are open to any discussion to improve our paper.
>
> Best wishes,
> The authors.

---

> > ### Comment · Reviewer_GmMP · 2024-11-25
> >
> > Thank you for the response, which clarifies some of my concerns. I have a few follow-up questions:
> >
> > Q1.  In Figure 8, what determines the width of the band ('secialization')?
> >
> > Q3. What kind of empirical observations were used to conclude "final layer biases often hold sufficient representational power to approximate gradient alignment across objectives"?
> >
> > Q4. It is unclear why [1] is strictly concurrent work since it seems to have been public before the submission deadline. It is unusual to omit comparing with prior work at least qualitatively which addresses the same problem this paper is trying to solve.

---

> > > ### Author Response · Authors · 2024-11-26
> > >
> > > Thanks for letting us know that our response resolves some concerns successfully! We appreciate the reviewer for the follow-up questions. Below, we provide detailed responses to each point.
> > >
> > > ---
> > >
> > > **Q1. Specialization in Figure 8**
> > > The width of the bands in Figure 8 represents the (normalized) accuracy of each model (solution) on every objective, as described in Appendix A.5.3. This metric reflects the degree of specialization for each solution on every objective. Wider bands indicate better specialization. MosT-trained solutions exhibit better diversity on specialization compared to baseline methods, leading to improved overall performance across all objectives, as shown in Figure 1.
> > >
> > > ---
> > >
> > > **Q3. Final Layer Bias and Representational Power**
> > > Our conclusion regarding final layer bias is based on the following empirical observations: we tested MosT using either all the parameters in the model or only the biases from the final layer. Both configurations delivered comparable performance. Hence, the final-layer biases alone are equally effective for gradient alignment across objectives, which leads to our claim.
> > >
> > > ---
> > >
> > > **Q5. Discussion of [1] as Concurrent Work**
> > > We acknowledge the concern regarding [1], which is another submission to ICLR 2025. While [1] was made public shortly before ICLR submission deadline, its timeline relative to our submission preparation makes it a concurrent work. Additionally, [1] primarily focuses on convex optimization problems and trains simple nonlinear models on synthetic datasets. This scope differs significantly from our work, which targets more complex, real-world scenarios. Due to these differences and the absence of open-source code for [1], direct experimental comparisons were not feasible. However, we recognize the value of addressing [1] qualitatively and have included a discussion in the revised manuscript (Section 2) to strengthen the context of our contributions.
> > >
> > > ---
> > >
> > > We hope these responses address your concerns. Please let us know if additional clarifications are needed.

---

> > > > ### Comment · Reviewer_GmMP · 2024-12-03
> > > >
> > > > Thank you for your response. Based on your response and other reviewers' comments, I will maintain my positive score.

---

### Official Review · Reviewer_RKfr · 2024-11-06

**Soundness:** 4
**Presentation:** 3
**Contribution:** 3
**Rating:** 6
**Confidence:** 4

**Summary:**

This paper introduces a framework called "Many-Objective Multi-Solution Transport" (MosT) aimed at efficiently optimizing numerous objectives across a limited set of models. Traditional multi-objective optimization methods struggle to balance many objectives simultaneously, often focusing on only a few, which can lead to neglected objectives or poor performance. MosT overcomes this limitation by assigning each model to a complementary subset of objectives, ensuring diverse solutions across the Pareto frontier. It does this by using a bi-level optimization process: (1) an optimal transport (OT) method matches objectives to solutions balancing the performance trade-offs (2) a multi-objective optimization (MOO) per solution to optimize assigned objectives using MGDA. The approach demonstrates success across applications such as federated learning, multi-task learning, and prompt learning in language models, outperforming simple baseline methods in terms of both accuracy and diversity.

**Strengths:**

- **Novelty for MOO:** The paper introduces a bi-level optimization method based on a very natural idea. Combining MGDA and OT is a elegant idea.

- **Specialized Model Expertise:** It's a keen observation to see that each model naturally becomes an "expert" for a specific subset of objectives as a result of sparse OT solution.

- **Applicable to multiple applications:** MOST demonstrates success across federated learning, multi-task learning, and mixture-of-prompt learning, highlighting the framework’s adaptability.

**Weaknesses:**

- **Missing stronger baselines**: The paper compared MOST against trivial benchmarks. If the goal is to compare methods based on their diversity and the coverage of Pareto Front, nontrivial benchmarks like [1] (and benchmarks therein) must be also taken into consideration.

- **Evaluation Metrics**: Hypervolume [2] has been the established metric for evaluating and measuring coverage in Pareto Front for MOO methods. The reviewer does not see average accuracy across all objectives in all models as a good metric.


[1] Ren, Yinuo, et al. "Multi-objective optimization via Wasserstein-Fisher-Rao gradient flow." International Conference on Artificial Intelligence and Statistics. PMLR, 2024.

[2] https://gist.github.com/silviutofan92/df50cce796f226c57e958a6fc744b9fb

**Questions:**

* The paper claims "We do not evaluate on hypevolume for many-objective scenarios due to its computational complexity and infeasibility in high-dimensional settings (i.e., many objectives)". Why? What's the time complexity of computing hypervolume of  M solutions over N objectives?

* In the chosen evaluation metric of average accuracy, what's the average over? Is it over all M solutions and all N objectives (average over MxN items)?

* In 3.2, the paper discusses the case of having more solutions than objective (M>N). What's the goal in here? We can clearly form "experts" i.e. each solution optimizes a single objective. Is our goal to capture M different trade-offs among N objectives even though M>>N?

**Details Of Ethics Concerns:**

No ethics concern

---

> ### Author Response · Authors · 2024-11-22
> **Response to Reviewer RKfr**
>
> Dear Reviewer RKfr,
>
> Thank you for your detailed review and for recognizing the novelty and adaptability of our proposed method. We are pleased to address the points you raised and provide additional clarifications.
>
> ---
>
> > **Q1. ... to compare methods based on their diversity and the coverage of Pareto Front, nontrivial benchmarks like [1] (and benchmarks therein) must be also taken into consideration.**
>
> We appreciate the relevance of WFR-GF and will cite it. However, a direct comparison is not appropriate due to fundamental differences in the focused problems and evaluation. MosT specifically targets many-objective optimization ($n \gg m$) in practical challenges, while WFR-GF is designed for traditional multi-objective problems with few objectives. Furthermore, MosT is evaluated on diverse, real-world benchmark tasks—including federated learning and multi-task learning—covering highly heterogeneous and scalable objectives. This broader benchmarking demonstrates MosT’s applicability to practical, high-dimensional scenarios, which are beyond the scope of WFR-GF's synthetic datasets and focus on low-dimensional Pareto front exploration.
>
> > **Q2. Clarification on not using hypervolume for evaluation**
>
> The computational complexity of hypervolume calculations becomes prohibitive in high-dimensional many-objective scenarios, limiting its feasibility for our experiments. Specifically, calculating hypervolume for $m$ solutions over $n$ objectives has exponential time complexity in $n$, making it unsuitable for our large-scale many-objective tasks. Instead, we use average accuracy as a more important metric to practitioners, which reflects a balance between computational efficiency and the ability to capture overall performance across objectives.
>
> > **Q3. Explanation of the chosen evaluation metric (average accuracy)**
>
> Average accuracy is calculated by averaging the accuracy across all objectives, ensuring a balanced evaluation across both high- and low-performing objectives.
>
> > **Q4. Goal for the case where $M > N$ (Section 3.2). Is our goal to capture M different trade-offs among N objectives even though M >> N?**
>
> Yes, exactly. When $M > N$, our goal is to capture diverse trade-offs among the $N$ objectives rather than simply assigning each solution to a single objective. An example is shown in Figure 4. By maintaining $M$ solutions, we ensure a richer representation of the Pareto frontier, offering greater flexibility to accommodate various trade-offs and adapt to different contexts. This is especially valuable in practical applications where capturing fine-grained, nuanced trade-offs is critical for robustness and adaptability.
>
> ---
>
> Thank you once again for your insightful feedback. Your comments have provided us with valuable directions for strengthening our paper.
>
> Best regards,
> The Authors
>
> [1] Ren, Yinuo, et al. "Multi-objective optimization via Wasserstein-Fisher-Rao gradient flow." International Conference on Artificial Intelligence and Statistics. PMLR, 2024.

---

> > ### Comment · Reviewer_RKfr · 2024-11-24
> > **Changing my score from 5 to 6**
> >
> > I changed my score from 5 to 6 mostly because the drawbacks are not too significant to affect the novelty of the approach: mixing OT and MGDA.
> >
> > > However, a direct comparison is not appropriate due to fundamental differences in the focused problems and evaluation. MosT specifically targets many-objective optimization (
> > ) in practical challenges, while WFR-GF is designed for traditional multi-objective problems with few objectives.
> >
> > I disagree. Even though motivated by traditional MOO, WFR-GF also offers **multiple** solutions (i.e. number of particles) for **multiple** objective. To me it looks like a good benchmark to compare against
> >
> > > Specifically, calculating hypervolume for m solutions over n objectives has exponential time complexity in n, making it unsuitable for our large-scale many-objective tasks.
> >
> > > Average accuracy is calculated by averaging the accuracy across all objectives, ensuring a balanced evaluation across both high- and low-performing objectives.
> >
> > Fair point. Hypervolume is computationally (too) expensive. But average accuracy does not capture diversity in solutions either. I wish the authors could do hyper-volume for cases with smaller $n$ number of objectives.

---

### Official Review · Reviewer_oKXp · 2024-11-07

**Soundness:** 2
**Presentation:** 2
**Contribution:** 2
**Rating:** 6
**Confidence:** 4

**Summary:**

This work proposes a novel Multi-objective multi-solution Transport (MosT) method to find a small set of diverse Pareto solutions for problems with a large number of objectives. MosT is formulated as a bi-level optimization problem, where the upper-level problem is to determine different objective weights for each solution via optimal transport (OT) based on their current performance, while the lower-level problem is to find a Pareto solution for each subproblem with weighted objectives. MosT can also be generalized to tackle multi-objective optimization problems with few objectives via random objective interpolation. This work also provides theoretical analysis to prove MosT can find a set of Pareto solutions via solving the bi-level optimization problem. Experiments show MosT can achieve promising performance on different synthetic and application problems.

**Strengths:**

+ This work is well-organized and easy to follow.

+ The *small solution set for a large number of objectives* setting is important for many real-world applications. This work is a timely contribution to an interesting yet under-explored research direction.

+ The proposed MosT method can achieve promising performance on different synthetic and application problems.

**Weaknesses:**

I was a reviewer of an earlier version of this work, and I have read the current submission in detail. I have the following major concerns about the proposed MosT method.

**1. Motivation**

The key motivation of this work is to propose "a framework that finds multiple diverse solutions in the Pareto front of many objectives" and "ensures convergence to Pareto stationary solutions for complementary subsets of objectives". However, the terms diversity and complementary subsets are not clearly motivated and discussed in this work. The informal definition of diversity in Section 4 is helpful but not enough. How does the diversity connect to the complementary subsets of solutions found by MosT? Can we clearly define the "complementary subset" for MosT? Will the optimal solution set of MosT optimize some concrete diversity metrics?

**2. Balanced Weight Matrix**

For the proposed optimal transport based solution-objective matching method, it is not very clear why we prefer a balanced weight matrix for the objectives (i.e., "no model dominating on most objectives"). For a real-life application, if most of the objectives do share similar or even nearly identical structures, it is expected that one single solution can solve them well at the same time.

**3. Theoretical Analysis**

In MosT, the MGDAs can reach any Pareto optimal solution of each weighted problem, which is exactly the same Pareto optimal set of the original problem. In other words, the current theoretical guarantee of MosT is not stronger than the original MGDA. The reason why MosT can be ensured to find a set of diverse Pareto solutions does not have proper theoretical support.

**4. Related Work**

In most experiments, the final metric for comparison is the mean of average performance across all objectives. Why not directly optimize this metric using a specially designed method like the sum-of-minimum optimization method [1]? It seems that the method in [1] can efficiently match each objective-solution pair via a clustering-like approach. What are the pros and cons of MosT compared with this method?

[1] Efficient Algorithms for Sum-of-Minimum Optimization, ICML 2024.

**5. Runtime Analysis**

For the runtime comparison in Table 4, it is counter-intuitive to see linear scalarization require more runtime than MGDA. What is the computational overhead of linear scalarization over MGDA?

**6. Experiment**

Comparison with the related sum-of-minimum optimization method [1] is needed.

**Questions:**

See weaknesses.

---

> ### Author Response · Authors · 2024-11-22
> **Response to Reviewer oKXp**
>
> Dear Reviewer oKXp,
>
> Thank you for your detailed feedback. We appreciate the chance to address your concerns and clarify our work.
>
> ---
> > **Q1. How does diversity relate to its complementary subsets? Can "complementary subsets" be clearly defined? Does MosT's solution set optimize specific diversity metrics?**
>
> The solutions of MosT are diverse since they focus on different subsets of objectives or different subregions on the Pareto front (PF), capturing different user preferences. On the other hand, the constraint in OT enforces all solutions together to cover all objectives or the whole PF, so they are complementary subsets. More details:
> - **Diversity** refers to the optimization of each solution focus on a different subset of objectives. We define diversity formally in Section 4 based on cosine similarity between solutions in terms of their matching weights to objectives in $\Gamma$. So it measures the distinctiveness of solutions' specialization on the objectives.
> - **Complementary subsets of objectives**: The OT constraints in Eq. 3 enforce a balanced assignment of $n$ objectives to the $m$ solutions. By further encouraging sparsity in the matching matrix $\Gamma$, **each solution in MosT optimizes a subset of objectives and all the solutions together cover all objectives.** So the subsets covered by different solutions are complementary.
> - **The theoretical guarantees** provided (Theorems 4.1 and 4.2) show that MosT encourages diverse solutions while ensuring convergence to Pareto-optimal points in both convex and non-convex settings.
>
> > **Q2. Why prefer a balanced weight matrix in solution-objective matching**
>
> A balanced weight matrix is essential for capturing diverse trade-off points on the Pareto front and addressing varied user preferences over objectives. While a single solution may suffice for highly similar objectives, MosT is designed for optimizing many diverse and often conflicting objectives. By balancing the weight matrix, MosT ensures that (1) no objective is ignored or downweighed; and (2) no solution is dominating others or over-specialized, providing broad and well-distributed coverage of the objective space. This approach allows MosT to meet a wide range of user preferences and promotes robustness by ensuring no single solution dominates.
> > **Q3. How is MosT theoretically distinct from MGDA**
>
> You are correct that for a single solution with fixed weights for objectives, MosT’s theoretical guarantee aligns with that of standard MGDA. However, MosT is designed to find a finite diverse set of multiple solutions, not just a single one. Through its bi-level optimization framework, MosT jointly optimizes the matching weights and solutions, leveraging optimal transport to dispatch objectives to solutions. This ensures each solution specializes in a complementary subset of objectives, promoting diversity and achieving a well-distributed Pareto set.
> > **Q4. Comparison with sum-of-minimum optimization method [1]**
>
> [1] is clustering-based, which has inherent limitations: a notorious one is solution collapse, i.e., only one solution gets fully trained and dominates others on all objectives, as shown in our analysis (Appendix A.5.1, Figure 7). Clustering approaches are also rigid, often failing in practical scenarios where cluster structure assumption does not hold, e.g., when task relationships are dynamic and cannot be neatly grouped. In Table 2, experiments with TAG (clustering-based) method on multi-task learning datasets show poor alignment between task groupings and model performance, achieving only 31.05% average accuracy on DomainNet, failing to outperform other baselines or MosT. These results emphasize that MosT’s general framework, which dynamically identifies task relationships and ensures Pareto optimality, is better suited for addressing practical challenges beyond the clustering setting.
>
> For your reference, we've provided the learnt task grouping results by TAG on DomainNet ($n = 6$, $m=4$):
>    - Group 1: [1,3]
>    - Group 2: [1,4,6]
>    - Group 3: [2,3]
>    - Group 4: [5,6]
>
> Furthermore, in the table below, we've included the model performance per task on the DomainNet dataset. It's evident that the model performance does not align with TAG's task grouping, possibly due to task conflicts leading to oversight of some objectives. This highlights the difficulty of identifying task groupings.
> |Task\Model|M1|M2|M3|M4|
> |-|-|-|-|-|
> |T1|**74.9%**|1.0%|74.7%|1.3%|
> |T2|33.7%|0.8%|0.2%|**34.0%**|
> |T3|0.4%|**70.0%**|69.4%|0.6%|
> |T4|1.1%|0.4%|**2.7%**|0.0%|
> |T5|0.4%|2.1%|0.0%|**7.6%**|
> |T6|0.8%|1.3%|0.4%|**3.2%**|
> > **Q5. Runtime analysis for linear scalarization vs MGDA**
>
> When the number of objectives is small, the computational cost of finding common descent directions in MGDA is negligible, so the observed runtime difference is likely due to other factors such as IO overheads or system latency.
>
> ---
> Thank you once again for your valuable insights and questions.
>
> Best,
> The Authors

---

> > ### Comment · Reviewer_oKXp · 2024-11-27
> >
> > Thank you for your response and new results, but some of the concerns have not been well addressed.
> >
> > **1. Theoretical Analysis and Diversity**
> >
> > It is still unclear to me how MosT can **theoretically** "ensure each solution specializes in a complementary subset of objectives, promoting diversity and achieving a well-distributed Pareto set". The current theorems only show the solutions found by MosT are Pareto optimal or Pareto stationary for the convex and non-convex settings, but do not require them to be well-distributed on the Pareto set. There is no theoretical guarantee on the relation (e.g., diversity) among the found solutions.
> >
> > **2. Discussion and Comparison with Related Work**
> >
> > The revised paper claims that "some clustering-based methods attempt to group objectives or tasks before optimization (Ding et al., 2024), but they rely on predefined groupings and lack the flexibility to dynamically adapt to nuanced relationships, often leading to suboptimal solutions." However, to my understanding, the method proposed by (Ding et al., 2024) indeed generalizes the **iterative** K-means++ to solve a similar problem considered in this work. In other words, it does not "group objectives or tasks before optimization" or "rely on predefined groupings". In fact, this method will **adaptively adjust the task grouping** at each iteration. Therefore, I believe its performance will be different from the TAG method provided in this paper.

---

> > > ### Author Response · Authors · 2024-11-28
> > >
> > > Thank you for your detailed feedback and for pointing out areas requiring further clarification. We address your comments below.
> > >
> > > ---
> > >
> > > **Q1. Theoretical Analysis and Diversity**
> > >
> > > We appreciate the reviewer’s concern regarding the theoretical guarantees for diversity. In our work, diversity refers to the specialization of each solution on distinct subsets of objectives. We formally define diversity in Section 4 using the cosine similarity between solutions, which measures how distinctly solutions align with objectives via their matching weights in $\Gamma$.
> > >
> > > The OT constraints in Eq. 3 enforce a balanced assignment of $n$ objectives to $m$ solutions, ensuring that all objectives are covered. Additionally, by encouraging sparsity in the matching matrix $\Gamma$, MosT ensures that each solution specializes in a subset of objectives, and these subsets are complementary across all solutions. This mechanism promotes diverse specialization of solutions.
> > >
> > > Theorems 4.1 and 4.2 provide theoretical guarantees that MosT converges to Pareto-optimal solutions in both convex and non-convex settings. While these theorems do not explicitly prove the well-distribution of solutions on the Pareto front, the balanced assignment and sparsity constraints implicitly promote diversity, as supported by our empirical results (Figure 4 and 5). We acknowledge this limitation and have clarified it in the revised paper, highlighting it as a direction for future work.
> > >
> > > **Q2. Discussion and Comparison with Ding et al., 2024**
> > >
> > > While (Ding et al., 2024) employs dynamic clustering by iteratively enhancing K-means++, it primarily targets clustering of data points. In contrast, we study the clustering of objectives in multi-objective optimization. Specifically, many objectives are "clustered" into a few solutions by OT in every iteration. Compared to data clustering, one unique challenge here is the cluster collapse in federated learning contexts, specifically, one solution performs better than all others on all objectives (all objectives collapse to one single cluster).
> > >
> > > This issue, as highlighted in personalization benchmarks (Yishay et al., 2020; Wu et al., 2022), severely limits the scalability and effectiveness of clustering methods when $k>2$. Our experimental results (Appendix A.5.1 and Figure 7) further validate this limitation (the 2nd and 3rd plots in Figure 7). These demonstrate that predefined clustering techniques, even if dynamic, fail to fully capture the nuanced conflicts and trade-off between objectives without collapsing into suboptimal groupings.
> > >
> > > ---
> > >
> > > [1] Shanshan Wu, Tian Li, Zachary Charles, Yu Xiao, Ziyu Liu, Zheng Xu, Virginia Smith. "Motley: Benchmarking heterogeneity and personalization in federated learning." arXiv preprint arXiv:2206.09262, 2022.
> > >
> > > [2] Mansour, Yishay, et al. "Three approaches for personalization with applications to federated learning." arXiv preprint arXiv:2002.10619 (2020).

---

> > > > ### Comment · Reviewer_oKXp · 2024-11-30
> > > >
> > > > Thank you very much for your multiple rounds of response for the previous and current submissions. I have also read other reviewers' comments and the corresponding responses.
> > > >
> > > > I still have some concerns about using MGDA-like algorithms to tackle problems with many objectives. However, I also do believe the small solution set for a large number of objectives setting is important for many real-world applications, and the proposed OT + MGDA method is novel. Therefore, I raise my score to 6.

---

> ### Author Response · Authors · 2024-11-25
> **Sincerely Looking Forward to Your Suggestions**
>
> Dear Reviewer,
>
> Given that the discussion period is coming to end soon, we sincerely look forward to your reply to our response, and we are open to any discussion to improve our paper.
>
> Best wishes,
> The authors.

---

### Official Review · Reviewer_z8sS · 2024-11-08

**Soundness:** 3
**Presentation:** 4
**Contribution:** 4
**Rating:** 8
**Confidence:** 4

**Summary:**

The paper proposes MoST, an approach based on optimal transport, that finds multiple Pareto solutions with many (can outnumber the solutions) objectives.

**Strengths:**

It is interesting to see how optimal transport can be applied in MOO, to find diverse solutions. And I find the application natural and sound.

This paper addresses a particular aspect in MOO where number of objectives can be much larger than the number of solutions, and still find 'diverse' solutions. To the best of my knowledge, this aspect is not previously addressed enough in modern MOO (e.g. gradient-based methods that can handle DNN training) context.

Experiments are quite comprehensive.

**Weaknesses:**

I don't find noticeable weakness in this paper.

**Questions:**

The convergence results seem to be regarding Pareto stationary only (which is fine). For convex and strongly-convex, is it possible to argue anything about Pareto optimality?

---

> ### Author Response · Authors · 2024-11-21
> **Response to Reviewer z8sS**
>
> Dear Reviewer z8sS,
>
> Thank you for your positive feedback and for recognizing the strengths and contributions of our work. We are pleased to provide clarification regarding your question.
>
> ---
>
> > **Q. Convergence results regarding Pareto stationary solutions and potential arguments for Pareto optimality in convex and strongly-convex settings**
>
> Thank you for raising this point. As shown in Proposition 1 of [1], when each loss function $\mathcal{L}_i(\theta)$ is convex and continuous in $\theta$, and the solution space $\Theta$ is convex, the Pareto frontier of the multi-objective optimization problem is also convex. This property allows us to guarantee that any Pareto stationary solution found by MosT is indeed Pareto optimal in these settings.
>
> Furthermore, in strongly-convex cases, the convergence guarantees provided in Theorem 2 (Strongly-Convex) in our paper ensure that the solutions converge to Pareto stationary points, and the convexity assumption ensures that these points are globally Pareto optimal.
>
>
> ---
>
> Thank you for your encouraging feedback and for highlighting this opportunity to improve the clarity of our theoretical results. We have incorporated these enhancements in the revised version.
>
> Best regards,
> The Authors
>
> [1] Wang, Yuyan, et al. "Small towers make big differences." arXiv preprint arXiv:2008.05808 (2020).

---

> > ### Comment · Reviewer_z8sS · 2024-11-28
> >
> > Thank you for the response.
> >
> > I appreciate the authors including additional discussion on this topic in the main body of the paper. One minor clarification: under convexity, Pareto Stationarity only implies weak Pareto optimality. Achieving Pareto optimality requires strict convexity.
> >
> > I have read the comments from other reviewers and found some of them to be quite insightful.
> >
> > That being said, given the novelty of this paper, particularly its attempt to combine OT with MGDA, I will maintain my score.

---

### Official Review · Reviewer_o2xh · 2024-11-12

**Soundness:** 3
**Presentation:** 3
**Contribution:** 2
**Rating:** 6
**Confidence:** 5

**Summary:**

This paper proposes a multi-objective multi-solution transport (MosT) framework to identify solutions that achieve diverse trade-offs across multiple objectives. The proposed method aims to generate solutions over the Pareto set of a multi-objective optimization (MOO) problem to maximize diversity, particularly in cases with a large number of objectives.

**Strengths:**

1. This work is well-organized and easy to follow. The idea is presented well.
2. The proposed method can achieve good performance on different synthetic and application problems.

**Weaknesses:**

1. When we just focus on the optimization problem Eq. (1-3), it is a multi-objective bi-level optimization problem. This problem is not new; many recent works have focused on designing gradient-based methods to solve it, as seen in [1] and [2]. Clearly, solving $\Gamma$ should rely on the current $\theta$ according to Eq. (2), so $\Gamma$ in the upper-level is a function of $\theta$, i..e. $\Gamma(\theta)$. So, when we try to calculate the gradient of the objective function in the upper-level w.r.t. $\theta$, there exists a hypergradient term $\frac{d\Gamma^*(\theta)}{d\theta}$. Can we just ignore the connection between $\Gamma$ and $\theta$ in the constraint part and directly write $\Gamma$ instead of $\Gamma(\theta)$ in the upper level? Can the author provide some related work to support their approach?

2. When the number of objectives is very large, I do not think MGDA-based methods can work well. The MGDA-based method needs to calculate the gradient of each objective in each iteration. This is not affordable when the model and the number of objectives are both large. I believe [3] can better solve such a problem, but the author does not discuss this work.

3. MGDA is a quite old baseline. I believe there are many MGDA-based methods, but the author does not discuss those works listed in [4].

4. MGDA and Linearization cannot reach the concave part in the Pareto front [4]. The author should discuss solution diversity with [4].

5. Why does TAG fail in the MTL setting? In my view, the TAG just groups the tasks and trains these groups individually. It should not fail in such a setting. I quite doubt the authors' results.

6. Does $\Delta$ represent simplex in this paper? If so, the authors should use $\Delta^{n-1}$ and $\Delta^{m-1}$ in this paper, since $n-$simplex is a subset of $\mathbb{R}^{n+1}$. The author can check the corresponding definition in Wikipedia: https://en.wikipedia.org/wiki/Simplex

7. The article seems to have some compilation errors, e.g. $\tau=0$ in Line 280 and different style dash in Line 74 and 86.

8. Does $|| a ||$ represent the $l_2$ norm in this paper? I do not find where the author defines it since the authors also use $|| a ||_2$ in Eq. (6).

9. I think the authors did not read their paper carefully. I see the expression "Eq. equation *" dozens of times in this paper (e.g., Line 196, 206), which I believe is an expression error.

10. Eq. (45-46) $\to$ Eq. (47) seems need $L_i$ to be bounded functions. The authors do not make this assumption.

11. Lines 190-215 are just MGDA. I think those expressions can be placed in the preliminary part.

---

[1] Ye et al. Multi-Objective Meta Learning, AIJ, 2024.

[2] Ye et al. A First-Order Multi-Gradient Algorithm for Multi-Objective Bi-Level Optimization, ECAI, 2024.

[3] Yu et al. Enhancing Meta Learning via Multi-Objective Soft Improvement Functions, ICLR 2023.

[4] Lin et al. Smooth Tchebycheff Scalarization for Multi-Objective Optimization, ICML, 2024.

**Questions:**

See my questions in the weakness part.

---

> ### Author Response · Authors · 2024-11-22
> **Response to Reviewer o2xh**
>
> Dear Reviewer o2xh,
>
> Thank you for your detailed review and constructive feedback. We have addressed each of your comments below and clarified specific aspects of our work.
>
> ---
> > **Q1. problem not new ... Clarification on bi-level optimization and the dependency between $\Gamma$ and $\theta$**
>
> MosT addresses a novel problem: optimizing a small, diverse set of solutions for many-objective scenarios ($n \gg m$). Unlike existing works that focus on finding a single Pareto solution, MosT aims at the diversity of multiple solutions and their coverage over all objectives, ensuring each solution specializes in complementary subsets. This approach is particularly relevant for practical cases with large numbers of objectives, where balancing trade-offs with limited solutions is crucial.
>
> We chose a simple connection between $\Gamma$ and $\theta$ in the upper-level optimization to avoid high-order gradients, which dominate computation and hinder efficiency. Estimating hypergradients $\frac{d\Gamma^*(\theta)}{d\theta}$ is computationally impractical and might not be reliable given the non-smooth nature of the optimal transport objective. Our simple design allows MosT to remain computationally efficient while effectively optimizing diverse solutions.
>
> > **Q2. Limitations of MGDA-based methods when the number of objectives is large**
>
> It is challenging to apply vanilla MGDA to many objectives. But a very aim of MosT is to address this limitation of MGDA by assigning objectives to different solutions to be optimized, by leveraging optimal transport (OT).
>
> As demonstrated in Proposition 1, Figure 2, and Section 5, OT induces sparsity in the solution-objective assignment matrix $\Gamma$, effectively reducing the number of competing objectives when optimizing each solution. This sparse grouping of objectives alleviates and avoids the well-known gradient conflicts in MGDA, especially on many objectives. By dynamically balancing objectives optimized for different solutions, MosT ensures more efficient optimization and better scalability, compared to standard MGDA-based approaches.
> > **Q3. MGDA as a baseline and missing related work. Need for discussing solution diversity and the Pareto front with respect to [4]**
>
> While [4] provides an efficient scalarization approach for finding single solutions on the Pareto front, MosT tackles a different and broader challenge: optimizing many objectives with a limited budget of diverse solutions to be optimized. Our framework explicitly addresses the need for coverage and diversity in the solution space, leveraging optimal transport to allocate objectives dynamically.
> > **Q4. Clarification on the TAG failure in MTL settings**
>
> TAG fails in MTL settings primarily due to the negative transfer within each group of the static clustering. This approach often forces tasks into rigid groups, resulting in shared parameters that degrade individual task performance. In DomainNet, for example, tasks come from diverse styles or domains (e.g., real images, sketches, paintings), making static clustering unsuitable. Task 1, involving real images, is fundamental and broadly relevant to other tasks, but TAG’s rigid grouping fails to capture these nuanced relationships. In contrast, MosT dynamically adapts to task relationships, effectively balancing trade-offs and avoiding the limitations of static clustering.
>
> For your reference, we've provided the learnt task grouping results by TAG on both datasets:
>
> - **Office-10 Dataset ($n = 4$, $m=3$):**
>    - Group 1: Tasks [1,2]
>    - Group 2: Tasks [1,3]
>    - Group 3: Tasks [1,4]
>
> - **DomainNet Dataset ($n = 6$, $m=4$):**
>    - Group 1: Tasks [1,3]
>    - Group 2: Tasks [1,4,6]
>    - Group 3: Tasks [2,3]
>    - Group 4: Tasks [5,6]
>
> Furthermore, in the table below, we've included the model performance per task on the DomainNet dataset. It's evident that the model performance does not align with TAG's task grouping, possibly due to task conflicts leading to oversight of some objectives. This highlights the difficulty of identifying task groupings.
> |Task\Model|Model 1|Model 2|Model 3|Model 4|
> |-|-|-|-|-|
> |Task 1|**74.90%**|0.95%|74.71%|1.33%|
> |Task 2|33.65%|0.76%|0.19%|**34.03%**|
> |Task 3|0.38%|**69.96%**|69.39%|0.57%|
> |Task 4|1.14%|0.38%|**2.66%**|0.00%|
> |Task 5|0.38%|2.09%|0.00%|**7.60%**|
> |Task 6|0.76%|1.33%|0.38%|**3.23%**|
> > **Q5. Eq. (45-46)->Eq. (47) seems need to be bounded functions. The authors do not make this assumption.**
>
> It actually requires the difference between loss at the initial point and the loss at the $T+1$'s around to be bounded. The dependence on this gap is standard in optimization, e.g., Theorem 1 in [1].
>
> ---
> Thank you once again for your valuable feedback and for identifying specific areas for improvement! We have carefully addressed your comments and incorporated them into the revised manuscript.
>
> Best regards,
> The Authors
>
> [1] "Convergence guarantees for a class of non-convex and non-smooth optimization problems." PMLR, 2018.

---

> ### Author Response · Authors · 2024-11-25
> **Sincerely Looking Forward to Your Suggestions**
>
> Dear Reviewer,
>
> Given that the discussion period is coming to end soon, we sincerely look forward to your reply to our response, and we are open to any discussion to improve our paper.
>
> Best wishes,
> The authors.

---

> ### Comment · Reviewer_o2xh · 2024-11-27
>
> Thanks for the author's reply. But these replies did not completely solve my problem.
>
> ---
>
> **R1**: I understand that the proposed method tries to obtain a diverse set of solutions for many objective scenarios. It is different from standard gradient-based MOO, which only finds one solution.
>
> However, my question is when we **only focus on the optimization problem and optimization algorithm**. This bi-level optimization (The author mentions it multiple times) in the paper is not correctly introduced and fully analyzed. Eqs. (1-3) is actually ill-posed. The vector $\Gamma$ is in the set related to $\theta$, so it is a function w.r.t $\theta$. This is why I ask the author to provide some related work to support their optimization problem's formulation. Can we just directly consider $\Gamma$ as a constant in the upper level? Imho, the original upper-level should be $\Gamma(\theta)$, then we can replace this function to the optimal value $\Gamma^*$, where $\Gamma^*$ is solved by OT solver. (we can consider the OT solver's solution to be accurate). Then we just ignore the high order gradient so the proposed method is reliable. So, removing a high-order gradient can affect the optimization, and the author needs to analyze whether this error can be bounded.
>
> The author does not provide any evidence to support their alternating optimization algorithm because they consider $\Gamma$ as a constant in the upper level at the first step, which I think needs many words to explain. I understand the author's approach, but I think these descriptions are not rigorous enough.
>
> ---
>
> **R**2-**R**3: I understand that the proposed method tries to solve a different problem. However, MGDA is still an old baseline. In the multi-task learning area, many new baselines only need $\mathcal{O}(1)$ computational cost and can achieve better performance compared with MGDA. The proposed method still needs to calculate the gradient for each task in each iteration. Considering that the improvement of the proposed method on the Office-Caltech10 and DomainNet is not large compared with MGDA, I still suggest the authors add more baselines. Also, in the FL setting, two baselines are old.
>
> ---
>
> **R**5: I think the authors' reply is not correct.
>
> I checked Theorem 1 in the provided paper ("Convergence guarantees for a class of non-convex and non-smooth optimization problems"). The Theorem 1 only requires that $F_1-F^*$ is bounded. This is reasonable because the initial point can be considered as a constant, and the global optimal is also a constant if the function has a minimizer.
>
> However, what the authors used in Eq. (45-46)->Eq. (47) is the difference between the loss at the initial point and the loss at the $T+1$'s is bounded. These two things are different. We can not guarantee the generated sequence of $F_T$ is bounded. I suggest the author read the paper "Mitigating Gradient Bias in Multi-objective Learning: A Provably Convergent Approach, ICLR 2023". This paper uses bounded assumption to bound equation 78.
>
> In fact, the assumption of bounded function is very important in gradient-based MOO. Many papers try to design an algorithm to remove this assumption, such as "Mitigating Gradient Bias in Multi-objective Learning: A Provably Convergent Approach, ICLR 2023". Therefore, I feel surprised that the author uses this property without assuming it.
>
> ---
>
> Overall, I am not satisfied with the mathematical writing of this article, and the theoretical assumption in this article is also incorrect. Although the problem solved is novel, there are relatively few and older baseline methods compared. I will keep my score.

---

> ### Author Response · Authors · 2024-11-28
>
> Thank you for your detailed feedback. We respectfully address your concerns below:
>
> ---
> **R1** Bi-level Optimization and Theoretical Rigor: **Our problem formulation is rigorous and your proposed formulation is consistent with Eq. (1)-(3).**
> - "Can we just directly consider $\Gamma$ as a constant in the upper level?" - $\Gamma$ is a constant in the upper level optimization as shown in Eq. (1).
> - "then we can replace this function to the optimal value $\Gamma^*$, where $\Gamma^*$ is solved by OT solver" - The $\Gamma^*$ you described is exactly what we defined in Eq. (2), which is the optimal solution achieved by OT.
> - "Then we just ignore the high order gradient so the proposed method is reliable" - We did ignore the high order gradient in Eq. (6) so our method is reliable.
>
> ---
> **R2-R3** Baselines: we proposed MosT as a general multi-objective multi-solution optimization that can address rich and diverse learning problems. We examined it on three real applications of three different machine learning tasks. Because **MosT is NOT specifically designed and optimized for one task (e.g., FL)**, we chose the most principal and representative approaches for each task as the baselines. MosT outperforms these baselines in every task on several benchmarks, validating its advantages and generalizability.
>
> ---
> **R5** Boundedness Assumptions: Eq. (45-46) -> Eq. (47) simply holds true since we have $L_i(\theta_j^1) -  L_i(\theta_j^{T+1}) \leq  L_i(\theta_j^1) -  L_i(\theta_j^*)$ by the definition of the minimizer $\theta_j^*$. So we do not need to assume the generated sequence is bounded. We understand your intuition but it might add an unexpected layer of complexity to the problem.
>
> ---
> We appreciate your thoughtful suggestions and will further clarify the above points in the next version to avoid any misunderstanding as yours. We hope we have clarified your confusion. Thanks!

---

> ### Comment · Reviewer_o2xh · 2024-11-29
>
> Thanks for the author's reply.
>
> ---
>
> **R1**: I think I have made my viewpoint on this problem very clear. **I'm not saying that the author's approach is wrong. What I'm saying is that this approach is not standard and requires evidence.**
>
> The challenge of bi-level optimization is the nested dependency between upper-level variables and lower-level variables. Most studies of bi-level optimization focus on addressing this challenge because this nested dependency leads to hypergradients. However, the author simply uses one sentence to solve this challenge: "At a high level, the bi-level optimization problem described above can be decoupled into two sub-problems (over Γ and θ1:m)when fixing one variable and optimizing the other." (Line 178). **No citation and no explanation.** After this sentence, the bi-level structure and hypergradient simply disappeared. The problem becomes a single-level MOO. My point is this is not a standard way to solve bi-level optimization problems. The Primary Area of this article includes "meta-learning". I think most representative methods in learn-to-learn (which can be considered as bi-level optimization), such as MAML and DARTS, do not totally ignore hypergradient terms similar to the approach in this paper.
>
> **If the authors think their approach is standard and the hypergradients can be removed in bi-level optimization without much impact, list several references and say we just follow their approach. Problem solved. If the authors think their approach is not standard and is new, analyze the effect of removing the hypergradient information. Problem solved. I think my question is not hard to answer, and both ways can enhance the optimization method.**
>
> In addition, I just want to correct one sentence in the authors' reply: "$\Gamma$ is a constant in the upper level optimization as shown in Eq. (1).". Once we write this nested optimization formulation (Eqs. (1-3)), $\Gamma$ naturally becomes a function w.r.t. $\theta$. This is determined by this optimized structure because changing $\theta$ will change the value of $\Gamma$. No matter how the author writes about $\Gamma$ in Eq. (1), it does not change this dependency. Only when we use a gradient-based method to solve this problem and focus on one iteration can we consider it as a constant (Like the authors claimed in Lines 198-201). Let's say for $t$-th iteration, we want to get $\theta_{t+1}$ with $\theta_{t}$ and calculate $\Gamma_t^*(\theta_t)$; in this case, we may decide whether to consider it as a constant. But, this is different from the original problem formulation (Eqs. (1-3)).
>
> ---
>
> **R5**: Thanks for your clarification. I find I misunderstood the authors' first reply, and the proof is correct. The problem here is different from my provided ref.
>
> ---
>
> I apologize to the author for my factual error and my incorrect judgment towards Q10. For this reason, I raised my score from 3 to 5. The author's reply does not change my concern about the rigor of the optimization algorithm and insufficient baseline methods.

---

> > ### Author Response · Authors · 2024-12-01
> >
> > Thank you for your feedback and suggestions!
> >
> > Our current theoretical analysis and algorithms are mainly based on the properties of alternating optimization. Since the low-level optimization (OT) cannot be solved by a gradient descent based method, existing hypergradient based bi-level optimization cannot be applied. This poses new challenges. So we will leave the development of hypergradient based algorithm and analysis to future work.
> >
> > We thank the reviewer for raising this point and will clarify it in the revised manuscript.

---

> > > ### Comment · Reviewer_o2xh · 2024-12-02
> > >
> > > Thank you very much for your multiple rounds of response. I have also read other reviewers' comments and the corresponding responses.
> > >
> > > I agree with other reviewer's view that OT + MGDA method is novel. However, I still have concerns about the rigor of the optimization algorithm and insufficient baseline methods. From my perspective, these are important problems. Judging everything plus other reviewers' comments, I think 5 rating is a fair score, so I stand at that.
> > >
> > > Thanks.

---

> > > > ### Author Response · Authors · 2024-12-03
> > > >
> > > > Thank you for your thoughtful and detailed response in the multi-round discussion! We appreciate your expertise, your acknowledgement of our novelty on proposing OT+MGDA and our method's good performance on different problems. We fully respect your concerns on the bi-level optimization formulation and baseline comparisons. In the following, we would like to briefly clarify and justify our choices of baselines:
> > > >
> > > > 1. **Difference with [1] on the number of solutions and the requirement of preference vector:**
> > > >    [1] focuses on optimizing a **single-solution** for multiple objectives and requires a predefined preference vector $\lambda$. In contrast, MosT jointly optimizes multiple solutions and dynamically balances their preferences (or assignments) to many objectives. So the preference of each solution in MosT is automatically adjusted in order to achieve a diverse yet complementary set of solutions on the Pareto front. This broader scope fundamentally differs from [1].
> > > >
> > > > 2. **Comparison with recent personalized FL methods:**
> > > >    Due to the multi-solution setting of MosT, a valid and fair comparison with personalized FL should focus on baselines that train multiple global models serving many clients (objectives). Hence, **the most related and recent FL baselines are clustering-based FL approaches.** As shown in Figure 7, the "objective selecting model" in the second plot is the IFCA algorithm proposed for clustering-based FL [2]. MosT (the leftmost plot) overcomes a notorious drawback of clustering-based FL (the 2nd and 3rd plot from the left), i.e., multiple clusters collapse to one as one global model outperforms all the others on all clients. In contrast, MosT achieves diverse global models with balanced coverage over all clients, which leads to the best performance on almost every client (as shown in Figure 1).
> > > >
> > > > We remain open to further discussion on the selection of baselines and their alignment with MosT's scope. Your feedback has been invaluable in improving our work, and we deeply appreciate your insights. Meanwhile, we would like to kindly ask you consider to raise the rating if you agree with any above arguments. Thanks!
> > > >
> > > > ---
> > > >
> > > > [1] Lin et al. Smooth Tchebycheff Scalarization for Multi-Objective Optimization, ICML, 2024.
> > > >
> > > > [2] Avishek Ghosh, Jichan Chung, Dong Yin, Kannan Ramchandran. "An Efficient Framework for Clustered Federated Learning." NeurIPS 2020.

---

### Meta-Review · Area_Chair_t96o · 2024-12-21

**Metareview:**

This work introduces a novel Multi-objective Multi-solution Transport (MosT) method designed to identify a compact set of diverse Pareto solutions for multi-objective optimization problems with a large number of objectives. MosT is formulated as a bi-level optimization problem: the upper-level problem determines objective weights for each solution using optimal transport (OT) based on their current performance, while the lower-level problem identifies a Pareto solution for each subproblem with weighted objectives.

Additionally, MosT can be generalized to handle multi-objective problems with fewer objectives by employing random objective interpolation. The paper provides theoretical analysis to demonstrate that MosT can reliably identify a set of Pareto solutions by solving the bi-level optimization problem.

This approach addresses a key challenge in multi-objective optimization, where the number of objectives significantly exceeds the number of solutions, and it ensures the discovery of diverse solutions. The proposed bi-level optimization framework is built on a straightforward yet effective idea, contributing to the advancement of methods for handling high-dimensional objective spaces.

**Additional Comments On Reviewer Discussion:**

The reviewers raised questions regarding the optimization problem, the proposed optimization algorithm, and its convergence properties. After multiple rounds of clarifications and responses, the reviewers acknowledged the novelty of the proposed approach.

---

### Decision · Program_Chairs · 2025-01-22

Accept (Poster)